**Wave spectral shapes in the coastal waters based on measured data off Karwar, west coast of**
**India**
Anjali Nair M,  Sanil Kumar V
Ocean Engineering Division
Council of Scientific & Industrial Research-National Institute of Oceanography
Dona Paula 403 004, Goa India
*Correspondence to email:sanil@nio.org Tel: 0091 832 2450 327
**Abstract**
Understanding of the wave spectral shapes is of primary importance for the design of
marine facilities. In this paper, the wave spectra collected from January 2011 to December 2015 in
the coastal waters of the eastern Arabian Sea using the moored directional waverider buoy are
examined to know the temporal variations in the wave spectral shape. Over an annual cycle, for
31.15% of the time, peak frequency is between 0.08 and 0.10 Hz and the significant wave height is
also relatively high (~ 1.55 m) for waves in this class. The slope of the high-frequency tail of the
monthly average wave spectra is high during the Indian summer monsoon period (June-September)
compared to other months and it increases with increase in significant wave height. There is not
much interannual variation in slope for swell dominated spectra during the monsoon, while in the
non-monsoon period, when wind-seas have much influence, the slope varies significantly. Since the
exponent of the high-frequency part of the wave spectrum is within the range from -4 to -3 during
the monsoon period, Donelan spectrum shows better fit for the high-frequency part of the wave
spectra in monsoon months compared to other months.
**Key Words**: Ocean surface waves, wind waves, Arabian Sea, wave spectrum, high-frequency tail

## 1. Introduction

Information on wave spectral shapes are required for designing marine structures (Chakrabarti, 2005) and almost all the wave parameters computations are based on the wave spectral function (Yuan and Huang, 2012). The growth of waves and the correspondent spectral shape is due to the complex ocean-atmosphere interactions, while the physics of air-sea interaction is not completely understood (Cavaleri et al., 2012). The shape of the wave spectrum depends on the factors governing the wave growth and decay, and a number of spectral shapes have been proposed in the past for different sea states (see Chakrabarti, 2005 for a review). The spectral shape is maintained by nonlinear transfer of energy through nonlinear four-wave interactions (quadruplet interactions) and white-capping (Gunson and Symonds, 2014). The momentum flux between the ocean and atmosphere govern the high-frequency wave components (Cavaleri et al., 2012). According to Philips, the equilibrium ranges for low-frequency and high-frequency region is proportional to $f^5$ and $f^4$ (where f is the frequency) respectively. Several field studies made since JONSWAP (Joint North Sea Wave Project) field campaign reveals an analytical form for wave spectra with the spectral tail proportional to $f^4$ (Toba, 1973; Kawai et al., 1977; Kahma, 1981; Forristall, 1981; Donelan et al., 1985). Usually, there is a predominance of swell fields in large oceanic areas, which is due to remote storms (Chen et al., 2002; Hwang et al., 2011; Semedo et al., 2011). The exponent used in the expression for the frequency tail has different values (see Siadatmousavi et al., 2012 for a brief review). For shallow water, Kitaigordskii et al. (1975) suggested $f^{-3}$ tail, Liu (1989) suggested $f^{-4}$ for growing young wind-seas and $f^{-3}$ for fully developed wave spectra. Badulin et al. (2007) suggested $f^4$ for frequencies where nonlinear interactions are dominant. The study carried out at Lake George by Young and Babanin (2006) revealed that in the frequency range $5f_p < f < 10f_p$, the average value of the exponent 'n' of $f^n$ is close to 4. Whereas, some studies in real sea conditions indicate that high-frequency shape of $f^4$ applies up to few times the peak frequency ($f_p$) and then decays faster with frequency. The spectra for coastlines in Currituck Sound with short fetch condition showed a decay closer to $f^5$ when f is greater than two or three times the peak frequency (Long and Resio, 2007). Gagnaire-Renou et al. (2010) found that the energy input from wind and dissipation due to white-capping have a significant influence on the high-frequency tail of the spectrum.

The physical processes in the north Indian Ocean have a distinct seasonal cycle (Shetye et
al., 1985; Ranjha et al., 2015) and the surface wind-wave field is no exception (Sanil Kumar et al.,
2012). In the eastern Arabian Sea (AS), significant wave height ($H_{m0}$) up to 6 m is measured in the
monsoon period (June to September), and during rest of the period, $H_{m0}$ is normally less than 1.5 m
(Sanil Kumar and Anand, 2004). Sanil Kumar et al. (2014) observed that in the eastern AS, the
wave spectral shapes are different at two locations within 350 km distance, even though the
difference in the integrated parameter like $H_{m0}$ is marginal. Dora and Sanil Kumar (2015) observed
that waves at 7-m water depth in the nearshore zone off Karwar are high energy waves in the
monsoon and low to moderate waves in the non-monsoon period (January to May and October to
December). Dora and Sanil Kumar (2015) study shows similar contribution of wind-seas and swells
during the pre-monsoon (February to May), while swells dominate the wind-sea in the post-
monsoon (October to January) and the monsoon period. A study was carried out by Glejin et al.
(2012) to find the variation in wave characteristics along the eastern AS and the influence of swells
in the nearshore waves at 3 locations during the monsoon period in 2010. This study shows that the
percentage of swells in the measured waves was 75% at the southern part of AS and 79% at the
northern part of AS. Wind and wave data measured at a few locations along the west coast of India
for short-period, one to two months as well as the wave model results were analysed to study the
wave characteristics in the deep as well as nearshore regions during different seasons (Vethamony
et al., 2013). From the wave data collected for two years period (2011 and 2012) along the eastern
AS, the swells of period more than 18 s and significant wave height less than 1 m which occur for
1.4 to 3.6% of the time were separated and their characteristics were studied by Glejin et al. (2016).
Anjali Nair and Sanil Kumar (2016) presented the daily, monthly, seasonal and annual variations in
the wave spectral characteristics for a location in the eastern AS and reported that over an annual
cycle, 29 % of the wave spectra are single-peaked spectra and 71 % are multi-peaked spectra.
Recently Amrutha et al. (2017) by analysing the measured wave data in October reported that the
high waves (significant wave height > 4 m) generated in an area bounded by 40-60° S and 20-40° E
in the south Indian Ocean reached the eastern AS in 5-6 days and resulted in the long-period waves.
The earlier studies indicate that the spectral tail of the high-frequency part shows large variation
and its variation with seasons are not known. Similarly, the shape of the parametric spectra are also
different and hence it is important to identify the spectral shapes based on the measured data
covering all the seasons and different years.

The discussion above shows that there is a strong inspiration to study the high-frequency
tail of the wave spectrum. For the present study, we used the directional waverider buoy measured
wave spectral data at 15-m water depth off Karwar, west coast of India, over 5 years during 2011 to
2015 and evaluated the nearshore wave spectral shapes in different months. This study addresses
two main questions: (1) How the high-frequency tail of the wave spectrum varies in different
months, and (2) What are the spectral parameters for the best-fit theoretical spectra.  This paper is
organized as follows: the study area is introduced in section 2, details of data used and
methodology in section 3. Section 4 presents the results of the study and the conclusions are given
in section 5.
**2.     Study area**

The coastline at Karwar is 24º inclined to the west from the north, and the 20 m depth
contour is inclined 29º to the west. Hence, large waves in the nearshore will have an incoming
direction close to 241º, since waves get aligned with the depth contour due to refraction. At 10, 30
and 75 km distance from Karwar, the depth contours of 20, 50 and 100 m are present (Fig. 1). The
study region is under the seasonally reversing monsoon winds, with winds from the northeast
during the post-monsoon and from the southwest during the monsoon period. The monsoon winds
are strong and the total seasonal rainfall is 280 cm. There is a 0.24 m annual cycle in mean sea level
from September to January.  The average tidal range is 1.58 m during spring tides and 0.72 m
during neap tides (Sanil Kumar et al., 2012).

**3.  Data and methods**

The waves off Karwar (14° 49' 56" N and 74° 6' 4" E) were measured using the directional
waverider buoy (DWR-MKIII) . Measurements are carried out from 1 January 2011 to 31
December 2015. The data of heave and two translational motion of the buoy are sampled at 3.84
Hz. A digital high-pass filter with a cut off at 30 s is applied to the 3.84 Hz samples. At the same
time it converts the sampling rate to 1.28 Hz and stores the time series data at 1.28 Hz. From the
time series data for 200s, the wave spectrum is obtained through a fast Fourier transform (FFT).
During half an hour 8 wave spectra of a 200 s data interval each are collected and averaged to get a
representative wave spectrum for half an hour (Datawell, 2009). The wave spectrum is with a
resolution of 0.005 Hz from 0.025 Hz to 0.1 Hz and is 0.01 Hz from 0.1 to 0.58 Hz. Bulk wave
parameters; significant wave height ($H_{m0}$) which equals $4\sqrt{m_o}$ and mean wave period ($T_{m02}$) based
on second order moment, which equals $\sqrt{m_0/m_2}$ ) are obtained from the spectral moments. Where
$m_n$ is the $n^{th}$ order spectral moment ( $m_n = \int_0^\infty f^n S(f) df$ , n=0 and 2), S(f) is the spectral energy
density and f is the frequency. The spectral peak period (Tp) is estimated from the wave spectrum
and the peak wave direction (Dp) is estimated based on circular moments (Kuik et al., 1988). The
wind-seas and swells are separated through the method described by Portilla et al. (2009) and the
wind-sea and the swell parameters are computed by integrating over the respective spectral parts.
Measurements reported here are in Coordinated Universal Time (UTC), which is 05:30 h behind
the local time. $U_{10}$ is the wind speed at 10-m height obtained from reanalysis data of zonal and
meridional components at 6 hourly intervals from NCEP / NCAR (Kalnay et.al., 1996) and is used
to study the influence of wind speed on the spectral shape.

Since the frequency bins over which the wave spectrum estimated is same in all years, the
monthly and seasonally averaged wave spectrum is computed by taking the average of the spectral
energy density at the respective frequencies of each spectrum over the specified time.

Wave spectrum continues to develop through non-linear wave-wave interactions even for
very long times and distances. Hence, most of the wave spectrum is not fully developed and cannot
be represented by Pierson-Moskowitz (PM) spectrum (Pierson and Moskowitz, 1964). Accordingly,
an additional factor was added to the PM spectrum in order to improve the fit to the measured
spectrum. The JONSWAP spectrum (Hasselmann et al., 1973) is thus a PM spectrum multiplied by
an extra peak enhancement factor $\gamma$. The high-frequency tail of the JONSWAP spectrum decays in
a form proportional to $f^{-5.}$ A number of studies reported that high-frequency decay is by a form
proportional to $f^{-4.}$ Modified JONSWAP spectrum including Toba's formulation of saturation range
was proposed by Donelan et al. (1985). The JONSWAP and Donelan spectrum used in the study
are given in eqns. (1) and (2).
$$S(f) = \frac{\alpha g^2}{(2\pi)^4 f^5} \exp\left[ -\frac{5}{4}\left(\frac{f}{f_p}\right)^{-4} \right] \gamma^{\exp\left[ -(f-f_p)^2 / 2\sigma^2 f_p \right]} \quad\quad\quad .............. (1)$$
$$S(f) = \frac{\alpha g^2}{(2\pi)^4 f^4 f_p} \exp\left[-\left(\frac{f}{f_p}\right)^{-4}\right] \gamma^{\exp\left[-(f-f_p)^2 \Big/ 2\sigma^2 f_p^{\,2}\right]}$$ ......................(2)
Where $\gamma$ is the peak enhancement parameter; $\alpha$ is Philip's constant; f is the wave frequency; g is the
gravitational acceleration and $\sigma$ is the width parameter.
$$\sigma = \begin{cases} 0.07, & f < f_p \\ 0.09, & f \geq f_p \end{cases}$$
An exponential curve $y = kf^b$ is fitted for high-frequency part of the spectrum and the
exponent (value of b) and the coefficient k is estimated for the best fitting curve based on statistical
measures such as least square error and bias. The slope of the high-frequency part of the wave
spectrum is represented by the exponent of the high-frequency tail.

For the present study, JONSWAP spectrum is tested by fitting for the whole frequency
range of the measured wave spectrum. It is found out that the JONSWAP spectra do not show a
good fit for higher frequency range, whereas Donelan spectrum shows better fit for the high-
frequency range. Hence, JONSWAP spectrum is used for the lower frequency range up to spectral
peak and Donelan spectrum is used for the higher frequency range from the spectral peak for
single-peaked wave spectrum. Theoretical wave spectra are not fitted to the double-peaked wave
spectra.

**4.    Results and discussions**

4.1    Bulk wave parameters

Mostly the wave conditions (~ 75%) at the buoy location are intermediate and shallow-
water waves (where water depth is less than half the wavelength, d < L/2), this condition is not
satisfied during ~ 25% of the time due to waves with mean periods of 4.4 s or less. This study,
therefore, deals with shallow, intermediate and deepwater wave climatology. Hence, bathymetry
will significantly influence the wave characteristics.

The persistent monsoon winds generate choppy seas with average wave heights of 2 m and

mean wave period of 6.5 s. Fig. 2 shows that in the monsoon, the observed waves had a maximum
$H_{m0}$ of about 5 m, with $H_{m0}$ of 2-2.5 m more common during this period. The maximum $H_{m0}$
measured during the study period is on 21 June 2015 17:30 UTC (Fig. 2a). Mean wave periods
($T_{m02}$) at the measurement location ranged from 4-8 s (Fig. 2b). Wave direction during monsoon is
predominantly from the west due to refraction towards the coast. The fluctuation in $H_{mo}$ due to the
southwest monsoon is seen in all the years (Fig. 2a). High waves ($H_{m0} > 2$ m) during 27-29
November 2011 are due to the deep depression ARB04 formed in the AS. During the study period,
the annual average $H_{m0}$ is same (~1.1 m) in all the years (Table 1). In 2013, the data during August
could not be collected and hence resulted in lower annual average $H_{m0}$. Over the 5 years, small
waves ($H_{m0} < 1$ m) account for a large proportion (63.94%) of measured data and only during
0.16% of the time, $H_{m0}$ exceeded 4 m (Table 2). The 25$^{th}$ and 75$^{th}$ percentiles of the $H_{m0}$ distribution
over the entire analysis period are 0.6 and 1.4 m.

Waves with low heights ($H_{m0} < 1$ m) are with the mean period in a large range (2.7-10.5 s),

whereas high waves ($H_{m0} > 3$m) have mean wave period in a narrow range (6.1-9.3 s) (Table 2).
For waves with $H_{m0}$ higher than 3 m, the Tp never exceeded 14.3 s and for waves with $H_{m0}$ less
than 1 m, Tp up to 22.2 s are observed (Fig. 2c) and the long period swells (14-20 s) are with $H_{m0} <$
2.5 m. Around 7% of the time during 2011-2015, waves have peak period more than 16.7 s (Table
3). Peak frequencies between 0.08 and 0.10 Hz, equivalent to a peak wave period of 10 - 12.5 s are
observed 31.15% of the time and the $H_{m0}$ is also relatively high (~ 1.55 m) for waves in this class.
During the annual cycle, the wave climate is dominated by low ($0.5 > H_{m0} > 1$m) intermediate-
period (Tp ~10-16s) south-westerly swell. Waves from the northwest are with Tp less than 8 s (Fig.

3).


The wave roses during 2011-2015 indicate that around 38% of the time during the period

2011 to 2015, the predominant wave direction is SSW (225°) with long period (14 - 18s) and
intermediate period (10 - 14s) waves (Fig. 3). A small percentage of long-period waves having $H_{m0}$
more than 1m are observed from the same direction in which more than 80% are swells (Fig. 3c).
Intermediate period waves observed having $H_{m0}$ less than 1m, contain 20 - 60% of swells. Around
10-15% of the waves observed during the period are from the west, which includes intermediate
and short period waves with $H_{m0}$ varying from 1.5 to 3m. These intermediate period waves from
west having $H_{m0}$ between 2.5 - 3m contain more than 80% of swells. Waves from NW are short
period waves with $H_{m0}$ between 0.5 and 1.5; in which swell percentage is very less showing the
influence of wind-sea (Fig. 3d). High waves observed in the study area consists of more than 80%
swells.

Date versus year plots of significant wave height (Fig. 4) shows that $H_{m0}$ has its maximum

values ($H_{m0}$ >3m) during the monsoon period with a wave direction of WSW and peak wave period
of 10 - 12s (intermediate period). The mean wave period shows its maximum values (6 - 8s) during
the monsoon period. During January–May in all the years, $H_{m0}$ is low ($H_{m0}$ < 1m) with waves from
SW, W and NW directions. NW waves observed are the result of strong sea breezes existing during
this period. Both long-period ($Tp$ > 14s), intermediate-period ($10 < Tp < 14s$) and short-period ($Tp$
< 8s) waves are observed during this period and hence, the mean wave period observed is low
compared to the monsoon (Fig. 4d). During October to December, similar to the pre-monsoon
period, $H_{m0}$ observed is less than 1m, but the wave direction is predominantly from SW and W, with
least NW waves. Short period waves are almost absent during this period, and the condition is
similar for all the years. The interannual variations in $H_{m0}$ are less than 15% (Fig. 4). Primary
seasonal variability in waves is due to the monsoonal wind reversal. During January-March, there is
a shift in the occurrences of northwest swells.

4.2    Wave spectrum

The normalized wave spectral energy density contours are presented for different years to

know the wind-sea/swell predominance (Fig. 5). Normalisation of the wave spectrum is done to
know the spread of energy in different frequencies. Since the range of maximum spectral energy
density in a year is large (~ 60 $m^2$/Hz), each wave spectrum is normalised through dividing the
spectral energy density by the maximum spectral energy density of that spectrum. The
predominance of both the wind-seas and swells are observed in the non-monsoon period, whereas
in the monsoon only swells are predominant (Fig. 5). The separation of swells and wind-seas
indicates that over an annual cycle, around 54% of the waves are swells. Glejin et al. (2012)
reported that the dominance of swells during monsoon is due to the fact that even though the wind
at the study region is strong during monsoon, the wind over the entire AS also will be strong and
when these swells are added to the wave system at the buoy location, the energy of the swell
increases (Donelan, 1987) and will result in dominance of swells. The spread of spectral energy to
higher frequencies (0.15 to 0.25 Hz) is predominant during January-May (Fig. 5) due to sea-breeze
in the pre-monsoon period (Neetu et al., 2006; Dora and Sanil Kumar, 2015). In the monsoon
during the wave growth period, the spectral peak shifts from 0.12-0.13 Hz to 0.07-0.09 Hz (lower
frequencies).

An interesting phenomenon is that the long-period (> 18 s) swells are present for 2.5% of
the time during the study period. The buoy location at 15 m water depth is exposed to waves from
northwest to south with the nearest landmass at ~ 1500 km in the northwest (Asia), ~ 2500 km in
the west (Africa), ~ 4000 km in the southwest (Africa) and ~ 9000 km in the south (Antarctica)
(Amrutha et al., 2017). Due to its exposure to the Southern Oceans and the large fetch available,
swells are present all year round in the study area and the swells are dominant in the non-monsoon
(Glejin et al., 2013). Throughout the year, waves with period more than 10 s (low-frequency < 0.1
Hz waves) are the southwest swells whereas with seasons the direction of short-period waves
changes (Fig. 5). Amrutha et al. (2017) reported that the long-period waves observed in the eastern
AS are the swells generated in the south Indian Ocean. In the monsoon season, the waves with
high-frequency are predominantly from west-southwest, whereas in the non-monsoon they are from
the northwest. In the non-monsoon period, the predominance of wind-seas and swells fluctuated
and hence the mean wave direction also changed frequently (Fig. 5). The average direction of
waves with $H_{m0}$ < 1m shows the northwest wind-seas and the southwest swells, whereas, for high
waves ($H_{m0}$ > 3m), the difference between the swell and wind-sea direction decreases. This is
because the high waves get aligned to the bottom contour before 15 m water depth on its approach
to the shallow water.

The interannual changes of wave spectral energy density for different months in the period
2011-2015 are studied by computing the monthly average wave spectra for all the years (Fig. 6). In
the non-monsoon period, the wave spectra observed is double-peaked, indicating the presence of
wind-seas and swells, whereas during the monsoon, due to the strong southwest winds, single
peaked spectrum is observed, i.e. the swell peak with low-frequency and high spectral energy
density. Along the Indian coast, Harish and Baba (1986), Rao and Baba (1996) and Sanil Kumar et
al. (2003) found out that wave spectra are generally multipeaked and that the double peaked wave
spectra are more frequent during low-sea states (Sanil Kumar et al., 2004). Sanil Kumar et al.
(2014), Sanil Kumar and Anjali (2015) and Anjali and Sanil Kumar (2016) have also observed that
double-peaked spectrum in the monsoon period in the eastern AS are due to the locally generated
wind-seas and the south Indian Ocean swells. In the study area, from January to May and October
to December, the swell peak is between the frequencies 0.07 and 0.08 Hz (12.5 < Tp < 14.3s), but
in the monsoon period, the swell peak is around 0.10 Hz, in all the years studied. This shows long-
period swells (Tp > 13s) in the non-monsoon period and intermediate period swells ( 8 < Tp <13s)
in the monsoon. Glejin et al. (2016), also observed the presence of low-amplitude long-period
waves in the eastern AS in the non-monsoon period and intermediate period waves in the monsoon
period. This is because of the propagation of swells from the southern hemisphere is more visible
during the non-monsoon period due to the calm conditions (low wind-seas) prevailing in the eastern
AS. Whereas during the monsoon period, these swells are less due to the turbulence in the north
Indian Ocean (Glejin et al., 2013). Large interannual variations are observed for monthly average
wave spectrum in all months except in July. This is because July is known to be the roughest month
over the entire annual cycle and southwest monsoon reaches its peak during July. Hence, the
influence of temporally varying wind-seas on the wave spectrum is least during July compared to
other months. Due to the early onset (on 1 June) and advancement of monsoon during 2013
compared to other years, the monthly average value of the maximum spectral energy is observed in
June 2013 (Fig. 6). The wave spectra of November 2011 is distinct from that of other years, with
low wind-sea peak frequency, i.e. 0.13 Hz due to the deep depression ARB04, occurred south of
India near Cape Comorin, during 26 November–1 December, with a sustained wind speed of 55
km/h. During October 2014, the second peak is observed at 0.11 Hz with comparatively high
energy showing the influence of cyclonic storm NILOFAR. It is an extremely severe cyclonic
storm that occurred during the period 25-31 October 2014, originated from a low-pressure area
between Indian and Arabian Peninsula, with the highest wind speed of 215 km/h and affected the
areas of India, Pakistan and Oman. Significant interannual variation is observed in the wind-sea
peak frequency. Wave spectra averaged over each season (Fig. 7) shows that the interannual
variations in energy spectra averaged over full year period almost follows the pattern of wave
spectra averaged over monsoon period, indicating the strong influence of monsoon winds over the
wave energy spectra in the study area. Interannual variations within the spectrum are more for
wind-sea region compared to swell region. During the study period, the maximum spectral energy
observed is during 2011 monsoon.

For different frequencies, the monthly average wave direction is shown in Fig. 8. It is
observed that throughout the year the mean wave direction of the swell peak is southwest (200-
250$^{\circ}$). In the non-monsoon period, the wind-sea direction is northwest (280-300°), except in
October and November. This is due to the wind-seas produced by sea breeze which has the
maximum intensity during the pre–monsoon season. Interannual variability in wave direction is
highest during October and November, where the wind-seas from southwest direction are also
observed. This is because, during these months, the wind speed and the strength of the monsoon
swell decreases, which makes the low energy wind-seas produced by the withdrawing monsoon
winds more visible.
Contour plots of spectral energy density (normalized) clearly show the predominance of
wind-seas and swells during the non-monsoon period (Fig. 9). Only Figs. 5 and 9 present the
normalised spectral energy density. In the monsoon period, the spectral energy density is mainly
confined to a narrow frequency range (0.07-0.14 Hz) and the wave spectra are mainly single peaked
with maximum energy within the frequency range 0.08-0.10 Hz, having direction 240$^{\circ}$. Glejin et al.
(2012) reported that in the monsoon season, the spectral peak is between 0.08 and 0.10 Hz (12-10s)
for ~ 72% of the time in the eastern AS. Earlier studies also reported dominance of swells in the
eastern AS during the monsoon (Sanil Kumar et al., 2012; Glejin et al., 2012). Above 0.15 Hz,
energy gradually decreases, with the lowest energy observed between 0.30 and 0.50 Hz. Wind-sea
energy is comparatively low during October, November and December and occurs mostly in the
frequency range less than 0.20 Hz, whereas, during January-May, the frequency exceeds 0.20 Hz.
In the pre-monsoon period, wind-sea plays a major role in nearshore wave environment (Rao and
Baba, 1996). Wind-sea energy is found to be low during April 2015 (Fig. 6), because of reduction
in local winds. The occurrence of wind-seas is very less during most of the time in November
except during 2011, due to the deep depression ARB04.
.
The behavior of the high-frequency part of the spectrum is governed by the energy balance
of waves generated by the local wind fields. When the wind blows over a long fetch or for a long
time, the wave energy for a given frequency reaches the equilibrium range and the energy input
from the wind is balanced by energy loss to lower frequencies and by wave breaking (Torsethaugen
and Haver, 2004). The high-frequency tail slope of the monthly average wave spectrum in different
years shows that the slope is high (b< -3.1), during June to September and the case is same for all
the years studied (Table 4). During all other months, the exponent in the expression for the
frequency tail is within the range - 3.1 to -1.5. The distribution of exponent values for different
significant wave height ranges shows that the slope increases (exponent decrease from -2.44 to -
4.20)  as the significant wave height increases and reaches a saturation range. For frequencies from
0.23 to 0.58 Hz in the eastern AS during January-May, Amrutha et al. (2017) observed that the
high-frequency tail has $f^{-2.5}$ pattern at 15 m water depth and for frequencies ranging from 0.31 to
0.55 Hz, the high-frequency tail follows $f^{-3}$ at 5 m water depth. Since $H_{mo}$ is maximum during the
monsoon period, the slope is also maximum during June to September. There is no much
interannual variation in slope for swell dominated spectra during the monsoon, while in the non-
monsoon period when wind-seas have much influence, the slope varies significantly.

The most obvious manifestations of nonlinearity are sharpening of the wave crests and the
flattening of the wave troughs and these effects are reflected in the skewness of the sea surface
elevation (Toffoli, 2006). Zero skewness indicates linear sea states, positive skewness value
indicate that the wave crests are bigger than the troughs. Figure 10 shows that nonlinearity
increases with increase in $H_{m0}$. The slope of the high-frequency end of the wave spectrum becomes
steeper when the wave nonlinearity increases. Donelan et al. (2012) find that in addition to the $k^{-4}$
dissipation that swells modulate the equilibrium in breaking waves dependent on the mean surface
slope, while Melville (1994) also quantified a relation between wave packet slopes and dissipation
rate. These results are specific to breaking waves, but one might expect similar relations between
surface dynamics and dissipation rate for non breaking waves. A function of the form: A * exp( $\lambda$
$H_{m0}$ ) + s0, with initial parameters of A = 8, $\lambda$ = -2.4, s0 = -3.7 is found to fit the exponent of the
high-frequency tail data with the significant wave height (Fig. 11a). The functional representation
of the exponent of the high-frequency tail data with $H_{m0}$ shown in Fig. 11a might be useful in
revealing the physical connection, and at the very least would provide a predictive basis relating
spectral slopes with mean significant wave heights as a basis for future research. It is shown in Fig.
11b that the exponent decreases (slope increases) as the mean wave period increases. The study
shows that the tail of the spectrum is influenced by the local wind conditions (Fig. 11c) and the
influence is more with the zonal component (u) of the wind than on the meridional component (v)
(Figs. 11e and 11f). The exponent of the high-frequency tail decreases with the increase of the
inverse wave age ($U_{10}/c$), where c is the celerity of the wave.

4.3       Comparison with theoretical wave spectra

In the monsoon period, the spectrum is single peaked with high spectral energy density and

during this period JONSWAP spectrum is fitted up to the peak frequency and after that Donelan
spectrum is used. The monthly average wave spectra during the monsoon period for the year 2011,
is compared with JONSWAP and Donelan theoretical wave spectra in Figure 12. It is found that
JONSWAP and Donelan spectra with modified parameters describe well the wave spectra at low
frequencies and high frequencies respectively. The values for $\alpha$ and $\Upsilon$ were varied from 0.0001 to
0.005 and 1.1 to 3.3 respectively to find out the values for which, the theoretical spectrum best fits
the measured spectrum and those values were used to plot the theoretical spectrum. The values of $\alpha$
and $\Upsilon$ thus obtained, for June, July, August and September are given in Table 6. From the table, the
average values of $\alpha$ and $\Upsilon$, for the monsoon months are obtained as 0.0009 and 1.82 for JONSWAP
spectra and 0.0274 and 1.64 for Donelan spectra respectively. These values are less than the
generally recommended values of $\alpha$ and $\Upsilon$; 0.0081 and 3.3. $\alpha$ is a constant that is related to the
wind speed and fetch length. For all the data, Donelan spectrum fitted is proportional to $f^n$, where n
is the exponent value of the high-frequency tail. The theoretical spectrum JONSWAP and Donelan
cannot completely describe the high-frequency tail of the measured spectrum since the high-
frequency tail in these spectrum decays in the form of $f^5$ and $f^4$ respectively. Since the exponent of
the high-frequency tail of the wave spectrum is within the range -4 to -3 during the monsoon
period, Donelan spectrum shows better fit for monsoon spectra compared to other months (Fig.

11).


**5.    Concluding remarks**

In this paper, the variations in the wave spectral shapes in different months for a nearshore

location are investigated, based on in situ wave data obtained from a moored directional waverider
buoy. Interannual variations within the spectrum are more for wind-seas compared to swells. The
maximum significant wave height measured at 15 m water depth is 5 m and the annual average $H_{m0}$
has similar value (~1.1 m) in all the years. Over the 5 years, small waves ($H_{m0} < 1$ m) account for a
large proportion of measured data (63.94% of the time). The study shows that high waves ($H_{m0} > 2$
m) are with spectral peak period between 8 and 14 s and the long period swells (14-20 s) are with
$H_{m0} < 2.5$ m. The high-frequency slope of the wave spectrum (the exponent decreases from -2.44 to
-4.20) increases with increase in significant wave height and mean wave period. During the
monsoon period, Donelan spectrum shows better fit for monsoon spectra compared to other months
since the exponent of the high-frequency part of the wave spectrum is within the range -4 to -3. The
decay of the high-frequency waves are fastest with depth and hence, the high-frequency tail values
observed in the study will be different for different water depths.

**Acknowledgments**
The authors acknowledge the Earth System Science Organization, Ministry of Earth
Sciences, New Delhi for providing the financial support to conduct part of this research. We thank
TM Balakrishnan Nair, Head OSISG and Arun Nherakkol, Scientist, INCOIS, Hyderabad and Jai
Singh, Technical Assistant, CSIR-NIO for the help during the collection of data. We thank Dr. Bhat
and Dr. J L Rathod, Department of Marine Biology, Karnataka University PG Centre, Karwar for
providing the logistics required for wave data collection. This work contributes part of the Ph.D.
work of the first author. This paper is dedicated to the memory of our esteemed colleague Ashok
Kumar, in recognition of his substantial contributions in initiating the long-term wave
measurements in the shallow waters around India. We thank the topic editor and both the reviewers
for their critical comments and the suggestions which improved the scientific content of the
publication. This publication is a NIO contribution.

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

**Figure captions**
Figure 1. Study area along with the wave measurement location in eastern Arabian Sea
Figure 2. Time series plot of a) significant wave height, b) mean wave period, c) peak wave period
and d) mean wave direction e) maximum spectral energy density from 1 January 2011 to 31
December 2015. Thick blue line indicates the monthly average values
Figure 3. Wave roses during 2011-2015 (a) significant wave height and mean wave direction, (b)
peak wave period and mean wave direction, (c) percentage of swell, (d) percentage of wind-sea and
mean wave direction
Figure 4. Date verses year plot of a) significant wave height b) mean wave direction, c) peak wave
period and d) mean wave period
Figure 5. Temporal variation of normalized spectral energy density (top panel) and mean wave
direction (bottom panel) with frequency in different years. The value used for normalizing the
spectral energy density is presented in Fig. 2e.
Figure 6. Monthly average wave spectra in 2011 to 2015
Figure 7. Wave spectra averaged over a) pre-monsoon (February-May), b) monsoon (June-
September), c) post-monsoon (October-January) and d) full year in different years
Figure 8. Monthly average wave direction at different frequencies in different months
Figure 9. Temporal variation of normalized spectral energy density in different months (data from
2011 to 2015 used). The value used for normalizing the spectral energy density is presented in Fig.
2e.
Figure 10. Scatter plot of significant wave height with skewness of the sea surface elevation in
different years
Figure 11.  Plot of exponent of the high-frequency tail with a) significant wave height, b) mean
wave period, c) wind speed, d) inverse wave age, e) u-wind and f) v-wind
Figure 12. Fitted theoretical spectra along with the monthly average wave spectra for different
month

Table 1. Number of data used in the study in different years along with range of significant wave
height and average value

| Year | Significant wave height (m) | | Number of data | %of data |
|---|---|---|---|---|
| | Range | Average | | |
| 2011 | 0.3-4.4 | 1.1 | 17517 | 99.98 |
| 2012 | 0.3-3.7 | 1.1 | 17323 | 98.61 |
| 2013 | 0.3-3.6 | 0.9[*] | 14531 | 82.94 |
| 2014 | 0.3-4.5 | 1.1 | 17284 | 98.65 |
| 2015 | 0.3-5.0 | 1.1 | 14772 | 84.32 |

[*] average value is estimated excluding the July month data
Table 2. Characteristics of waves in different range of significant wave height

| Significant wave height range | Number (percentage) | Range of Tp (s) | Mean Tp (s) | Range of $T_{m02}$ (s) | Mean $T_{m02}$ (s) |
|---|---|---|---|---|---|
| $H_{m0}$ < 1 m | 52062 (63.94) | 2.6-22.2 | 12.2 | 2.7-10.5 | 4.9 |
| $1 \leq H_{m0}$< 2 m | 18297 (22.47) | 3.6-22.2 | 10.5 | 3.4-10.7 | 5.7 |
| $2 \leq H_{m0}$< 3 m | 9839 (12.08) | 6.2-18.0 | 10.8 | 5.0-8.9 | 6.5 |
| $3 \leq H_{m0}$< 4 m | 1096 (1.35) | 10.0-14.3 | 11.8 | 6.1-9.1 | 7.2 |
| 4 m $\leq H_{m0}$ | 133 (0.16) | 10.5-14.3 | 12.6 | 7.2-9.3 | 7.8 |

Table 3. Average wave parameters and number of data in different spectral peak frequencies

| Frequency ($f_p$) range (Hz) | Number of data and % | $H_{m0}$ (m) | $T_{m02}$ (s) | Peak wave period (s) |
|---|---|---|---|---|
| $0.04 < f_p \leq 0.05$ | 318 (0.39) | 0.73 | 5.24 | 20.19 |
| $0.05 < f_p \leq 0.06$ | 5341 (6.56) | 0.82 | 5.48 | 17.16 |
| $0.06 < f_p \leq 0.07$ | 14764 (18.13) | 0.75 | 5.22 | 14.73 |
| $0.07 < f_p \leq 0.08$ | 18221 (22.38) | 0.80 | 5.05 | 12.96 |
| $0.08 < f_p \leq 0.10$ | 25364 (31.15) | 1.55 | 5.76 | 10.88 |
| $0.10 < f_p \leq 0.15$ | 9459 (11.62) | 1.25 | 5.35 | 8.07 |
| $0.15 < f_p \leq 0.20$ | 6355 (7.80) | 0.76 | 4.43 | 5.72 |
| $0.20 < f_p \leq 0.30$ | 1487 (1.83) | 0.78 | 3.86 | 4.36 |
| $0.30 < f_p \leq 0.50$ | 118 (0.14) | 0.66 | 3.22 | 3.09 |


Table 4. Exponent of the high-frequency tail of the monthly average wave spectra in different
years

| Months | Exponent of the high-frequency tail | | | | | |
|---|---|---|---|---|---|---|
| | 2011 | 2012 | 2013 | 2014 | 2015 | 2011-2015 |
| January | -2.08 | -2.93 | -2.97 | -2.72 | -2.81 | -2.72 |
| February | -2.41 | -3.02 | -2.74 | -2.99 | -3.06 | -2.85 |
| March | -2.75 | -2.91 | -2.82 | -2.76 | No data | -2.81 |
| April | -2.56 | -2.74 | -2.64 | -2.71 | -2.19 | -2.60 |
| May | -2.59 | -2.67 | -2.63 | -2.42 | -2.51 | -2.56 |
| June | -3.64 | -3.53 | -3.55 | -3.82 | -3.58 | -3.55 |
| July | -3.76 | -3.55 | No data | -3.82 | -3.63 | -3.70 |
| August | -3.63 | -3.58 | -3.40 | -3.52 | -3.65 | -3.58 |
| September | -3.41 | -3.44 | -3.16 | -3.38 | -3.00 | -3.30 |
| October | -2.02 | -2.77 | -3.03 | -2.52 | -2.61 | -2.68 |
| November | -1.78 | -2.43 | -1.77 | -1.55 | -1.65 | -1.84 |
| December | -1.69 | -2.23 | -1.95 | -2.06 | -1.79 | -1.94 |

Table 5. Exponent of the high-frequency tail of the average wave spectra in different wave height
ranges

| Range of $H_{m0}$ (m) | Exponent of the high-frequency tail |
|---|---|
| 0-1 | -2.44 |
| 1-2 | -3.26 |
| 2-3 | -3.67 |
| 3-4 | -4.21 |
| 4-5 | -4.21 |


Table 6. Parameters of the fitted wave spectrum in different years

| Year | | JONSWAP spectrum | | Donelan spectrum | |
|---|---|---|---|---|---|
| | | α | ϒ | α | ϒ |
| 2011 | June | 0.0013 | 2.2 | 0.0028 | 2.0 |
| | July | 0.0016 | 1.5 | 0.0021 | 1.7 |
| | August | 0.0013 | 1.8 | 0.0029 | 1.7 |
| | September | 0.0004 | 2.3 | 0.0021 | 1.6 |
| 2012 | June | 0.0015 | 1.6 | 0.0029 | 2.0 |
| | July | 0.0010 | 2.1 | 0.0031 | 1.9 |
| | August | 0.0009 | 2.2 | 0.0032 | 1.7 |
| | September | 0.0006 | 2.0 | 0.0024 | 1.8 |
| 2013 | June | 0.0006 | 3.3 | 0.0030 | 1.9 |
| | July | | | No data | |
| | August | 0.0012 | 1.1 | 0.0038 | 1.4 |
| | September | 0.0005 | 1.9 | 0.0042 | 1.4 |
| 2014 | June | 0.0010 | 1.1 | 0.0010 | 1.6 |
| | July | 0.0006 | 2.5 | 0.0019 | 1.2 |
| | August | 0.0006 | 1.5 | 0.0021 | 1.2 |
| | September | 0.0011 | 1.1 | 0.0032 | 1.4 |
| 2015 | June | 0.0011 | 1.4 | 0.0023 | 1.8 |
| | July | 0.0011 | 1.9 | 0.0024 | 1.8 |
| | August | 0.0008 | 1.8 | 0.0024 | 1.4 |
| | September | 0.0006 | 1.3 | 0.0043 | 1.6 |


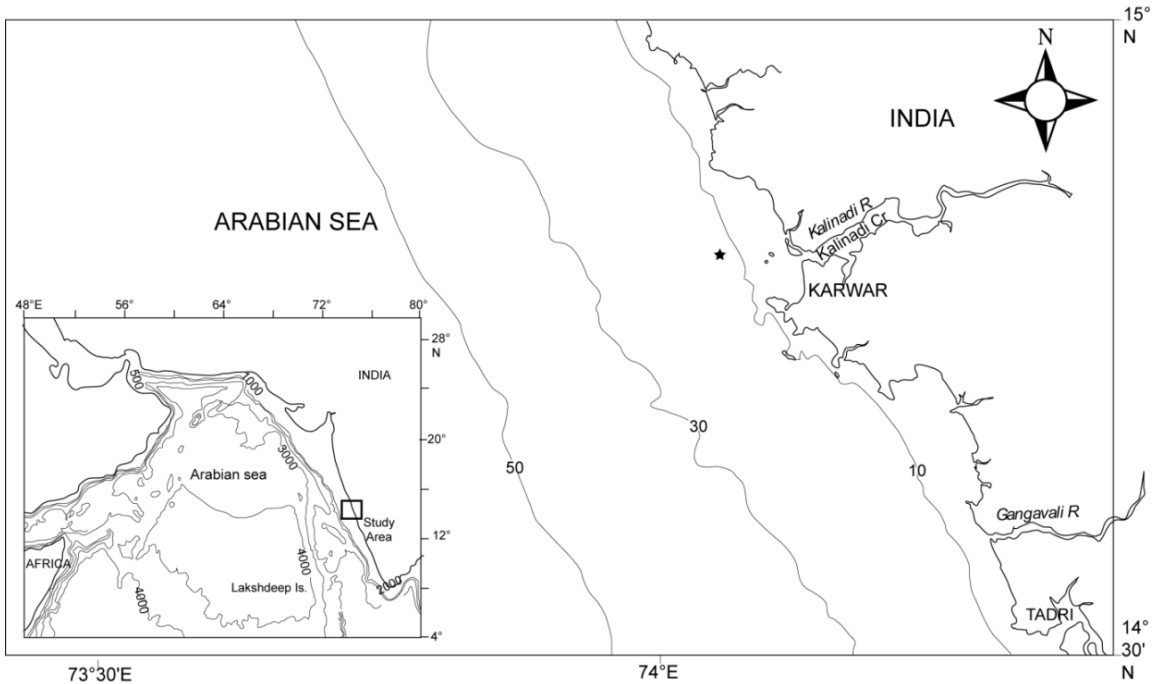

Figure 1. Study area along with the wave measurement location in eastern Arabian Sea

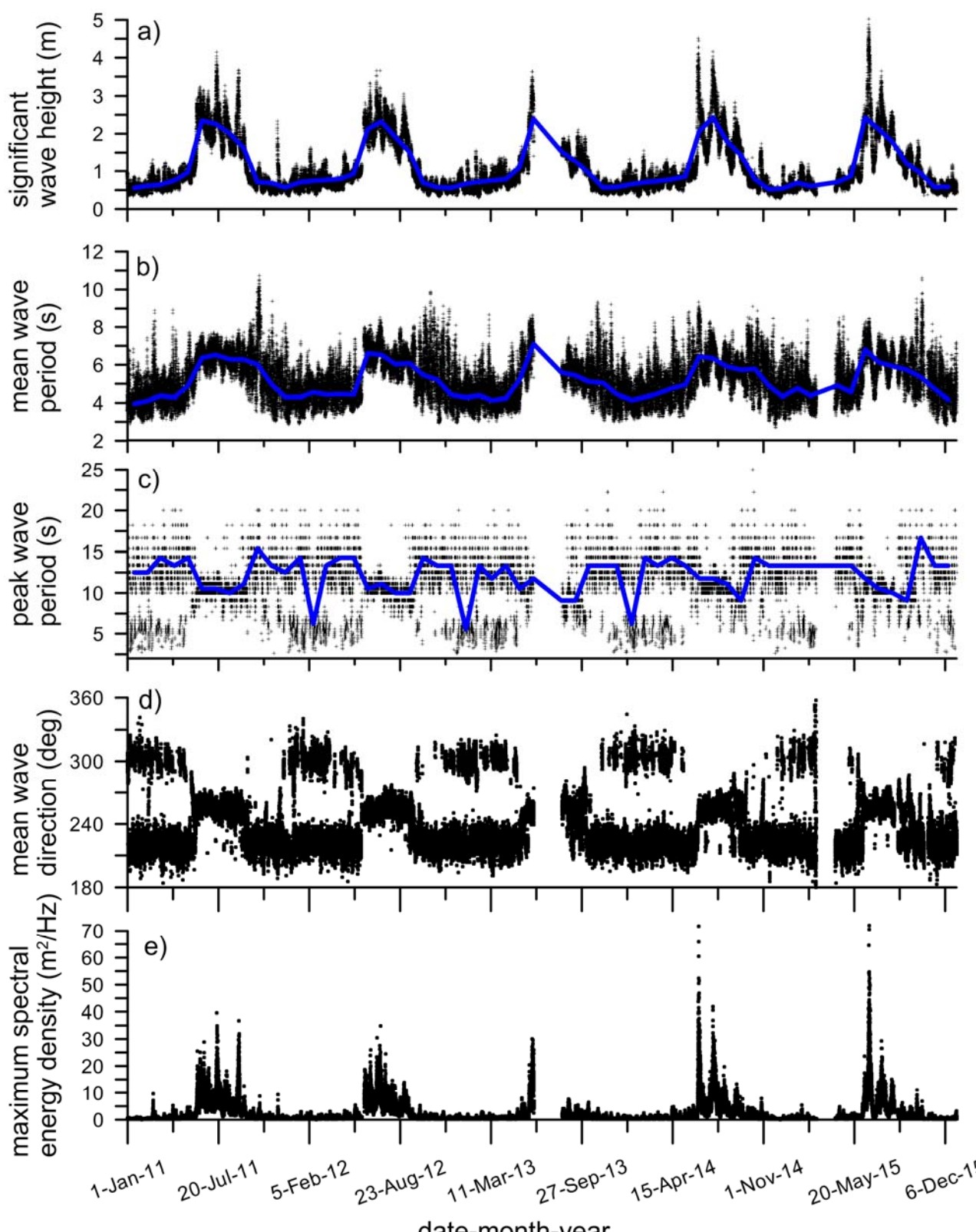

Figure 2. Time series plot of a) significant wave height, b) mean wave period, c) peak wave period
and d) mean wave direction e) maximum spectral energy density from 1 January 2011 to 31
December 2015. Thick blue line indicates the monthly average values

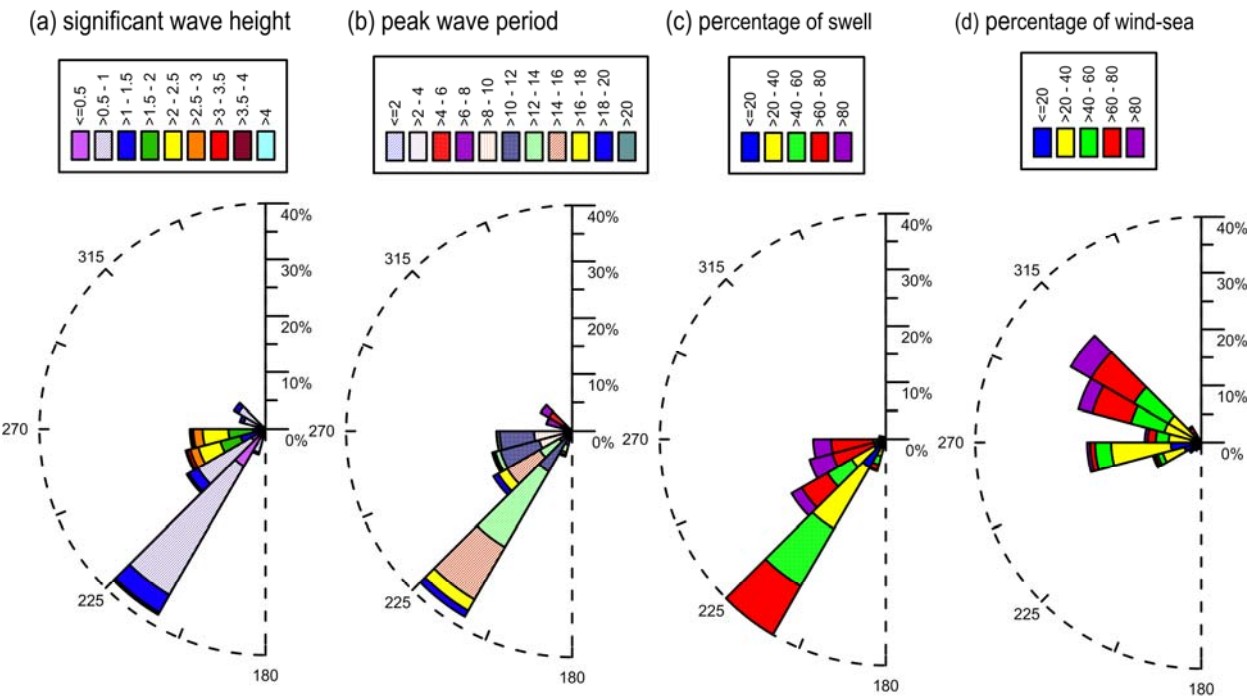

Figure 3. Wave roses during 2011-2015 (a) significant wave height and mean wave direction, (b)
peak wave period and mean wave direction, (c) percentage of swell, (d) percentage of wind-sea and
mean wave direction


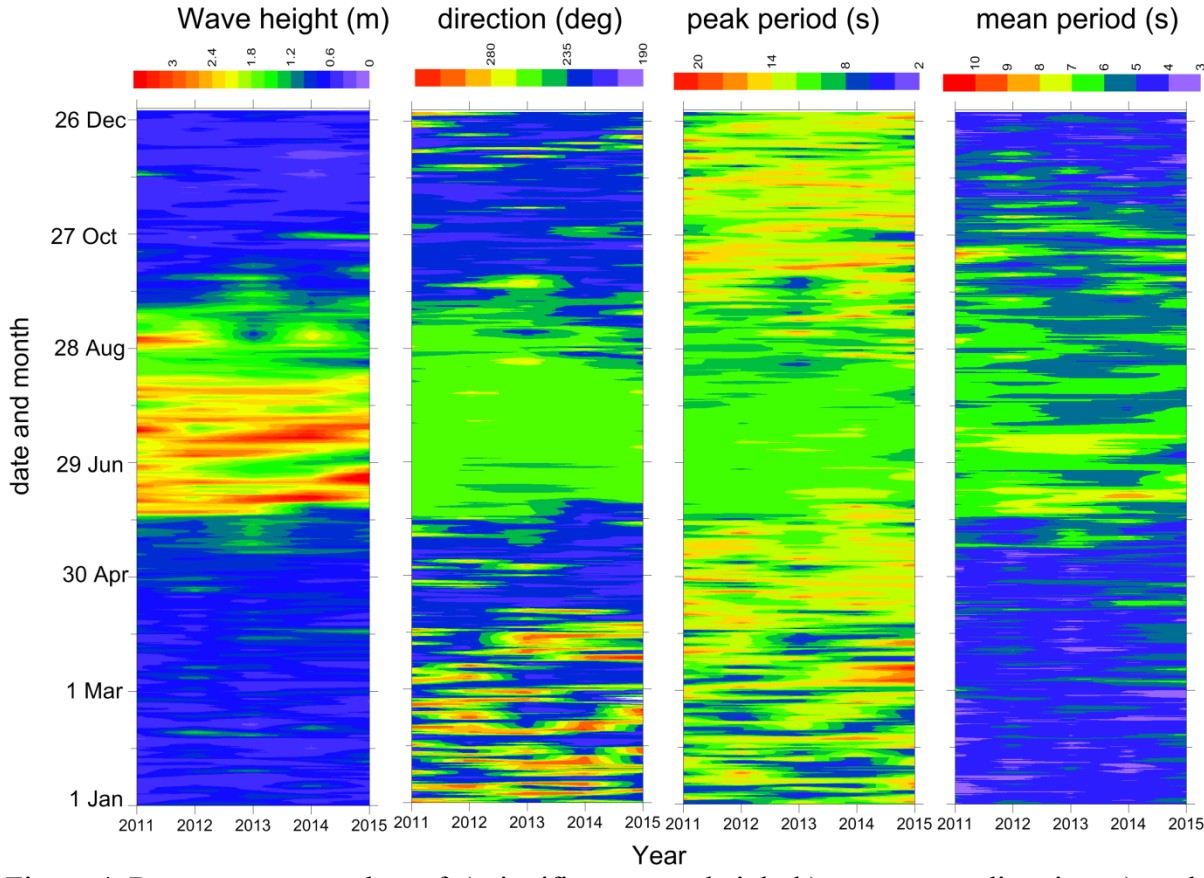

Figure 4. Date verses year plots of a) significant wave height b) mean wave direction, c) peak wave
period and d) mean wave period.



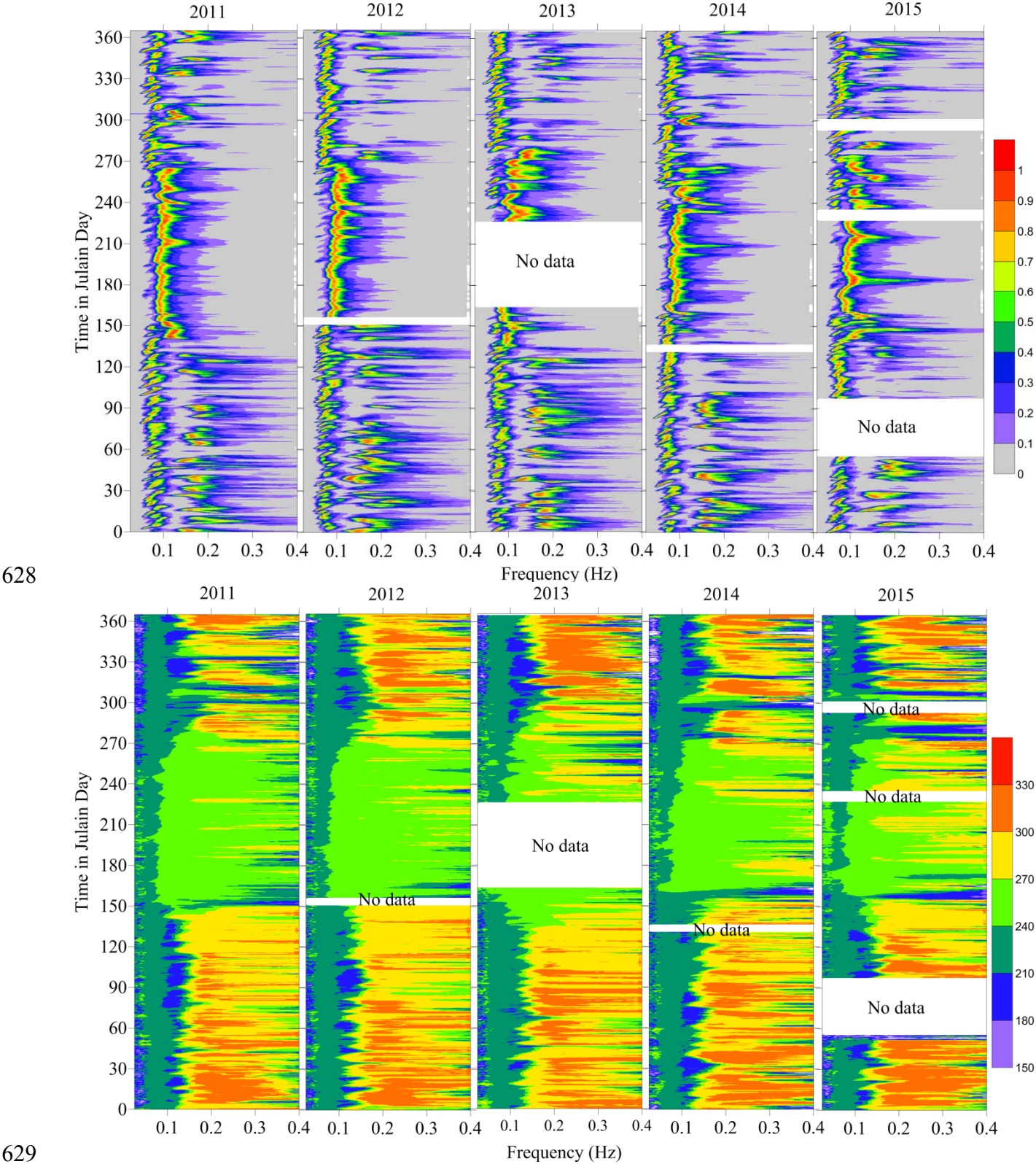



Figure 5. Temporal variation of normalized spectral energy density (top panel) and mean wave
direction (bottom panel) with frequency in different years. The value used for normalizing the
spectral energy density is presented in Fig. 2e.

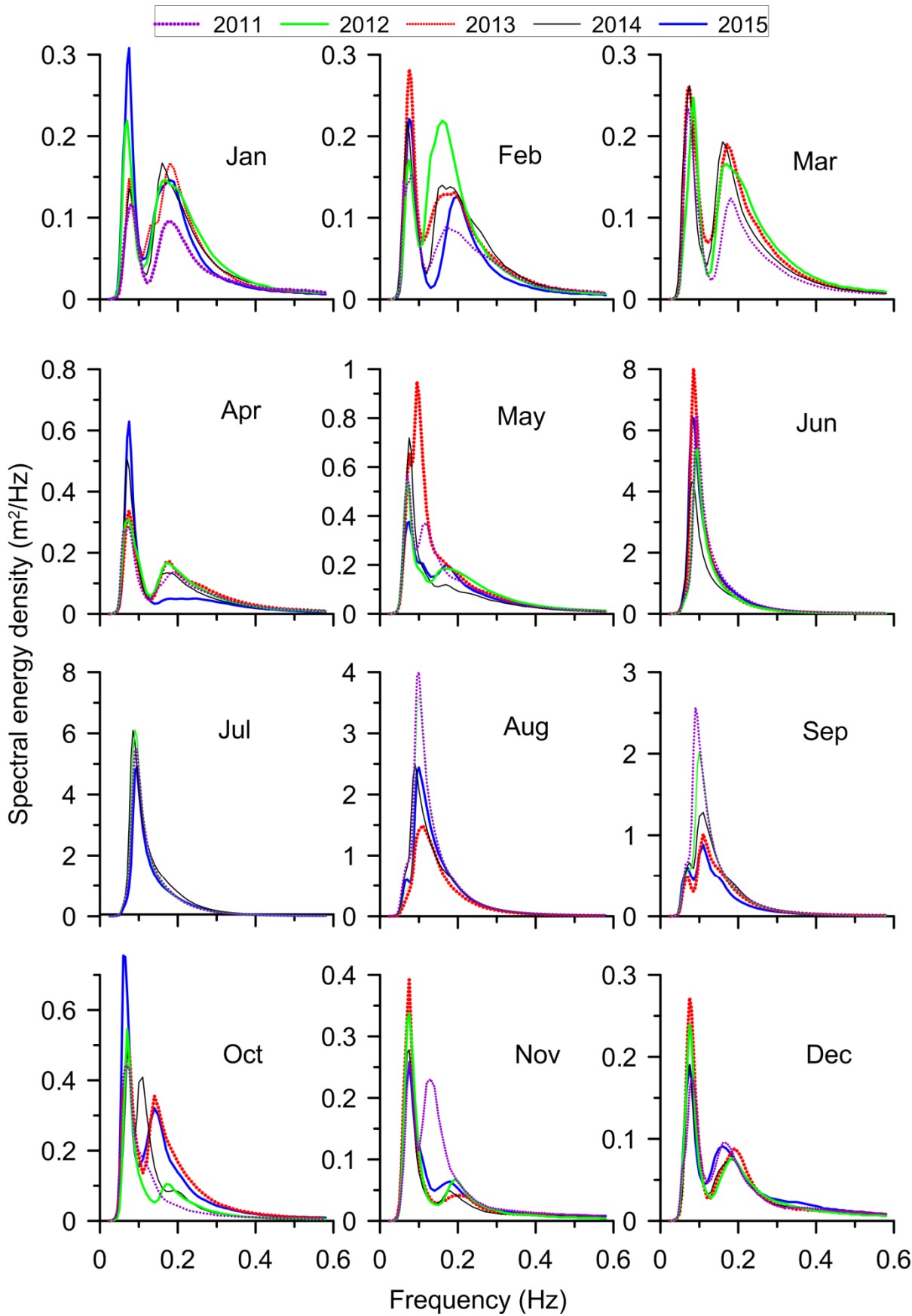

Figure 6. Monthly average wave spectra in 2011 to 2015

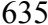

Figure 7. Wave spectra averaged over a) pre-monsoon (February-May), b) monsoon (June-
September), c) post-monsoon (October-January) and d) full year in different years

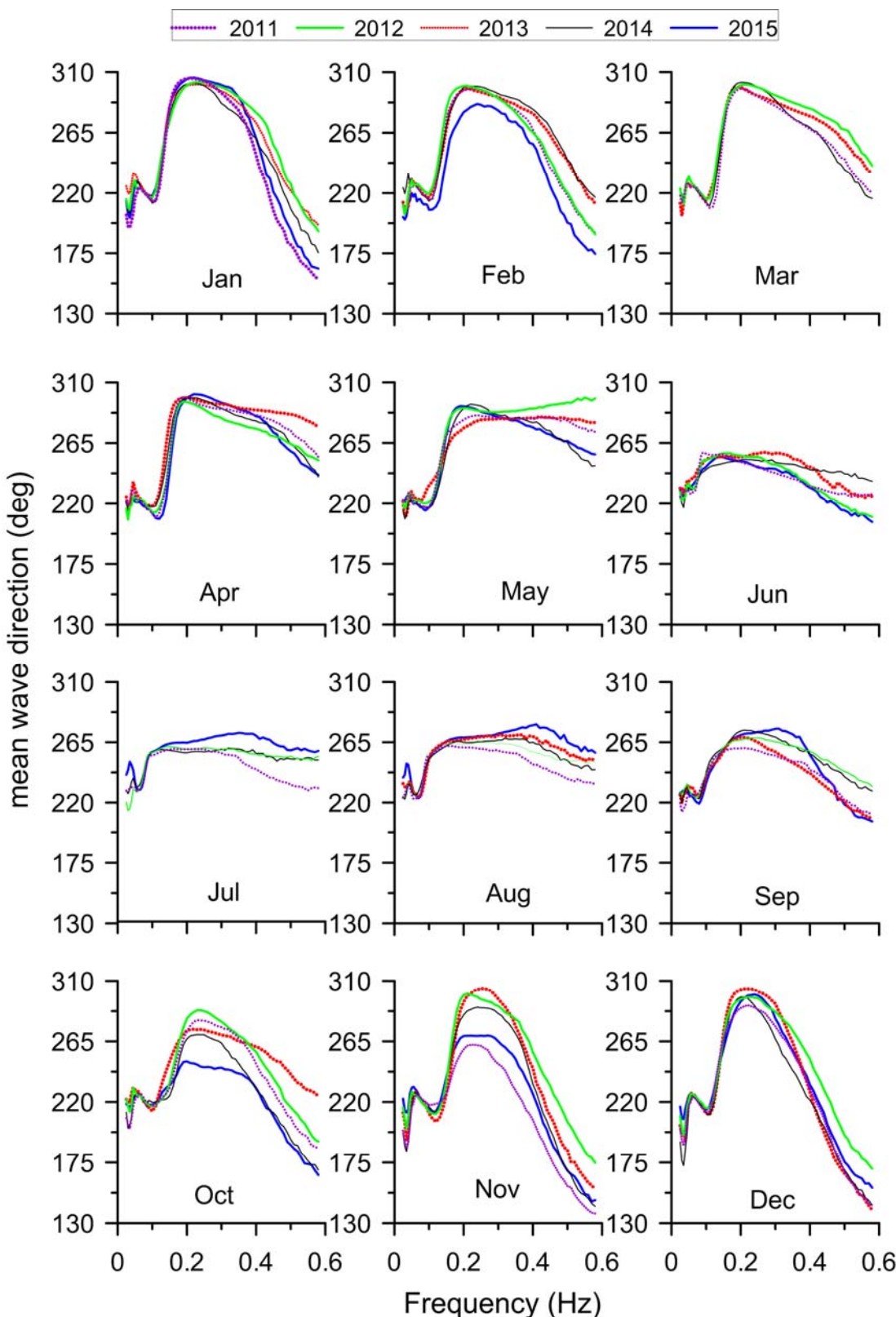

Figure 8. Monthly average wave direction at different frequencies in different months

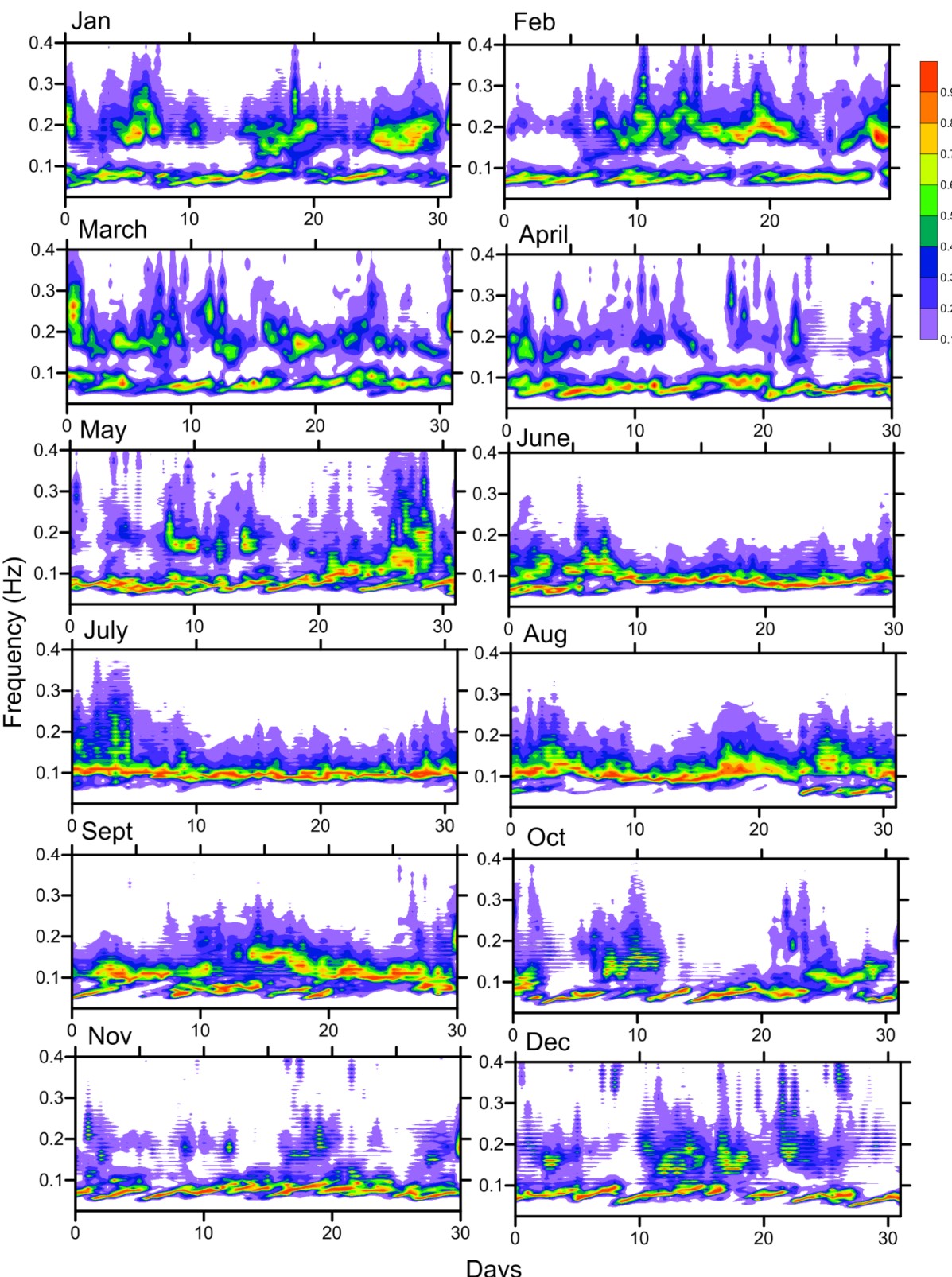

Figure 9. Temporal variation of normalized spectral energy density in different months (data from 2011 to 2015 used). The value used for normalizing the spectral energy density is presented in Fig. 2e.

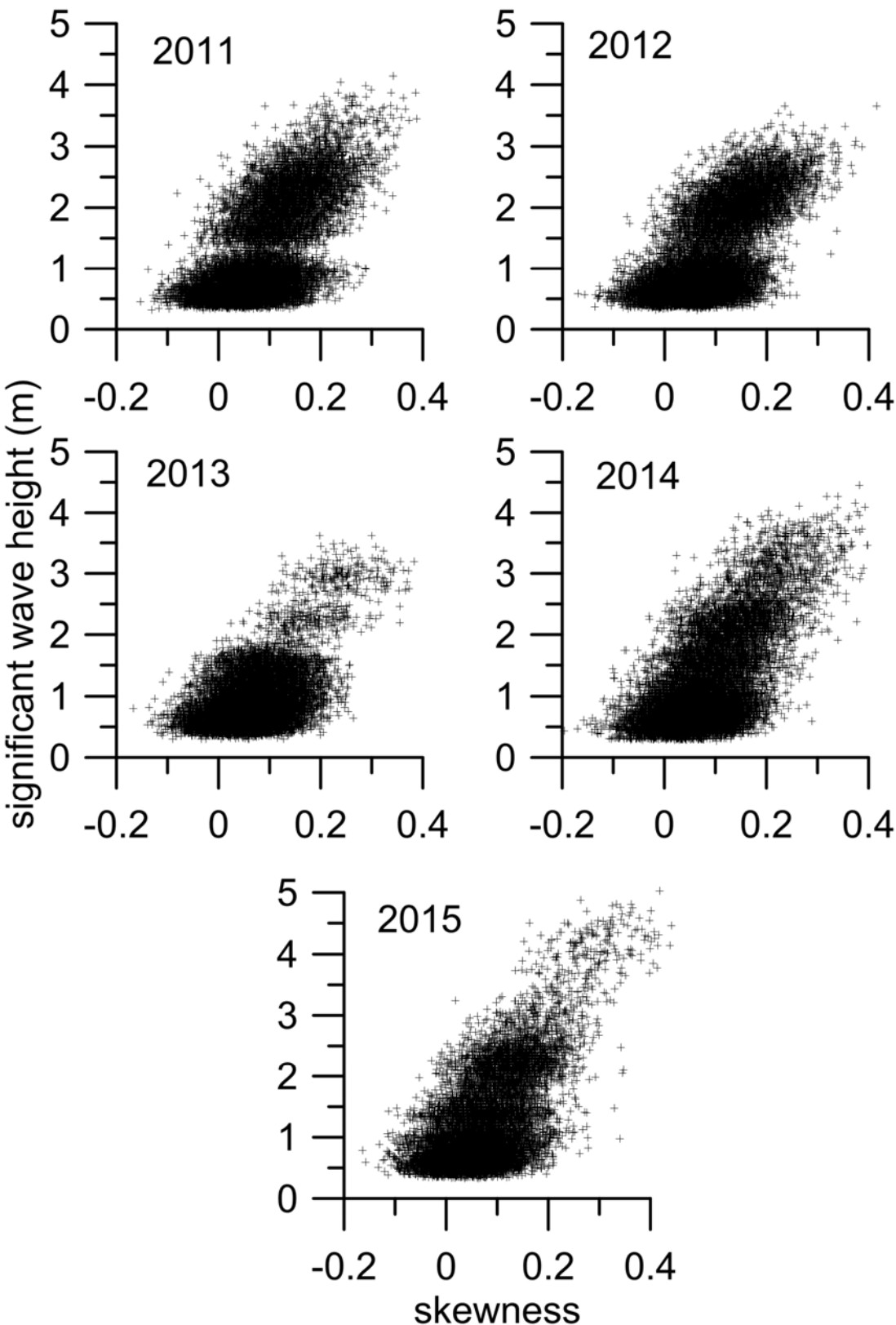



Figure 10. Scatter plot of significant wave height with skewness of the sea surface elevation in
different years

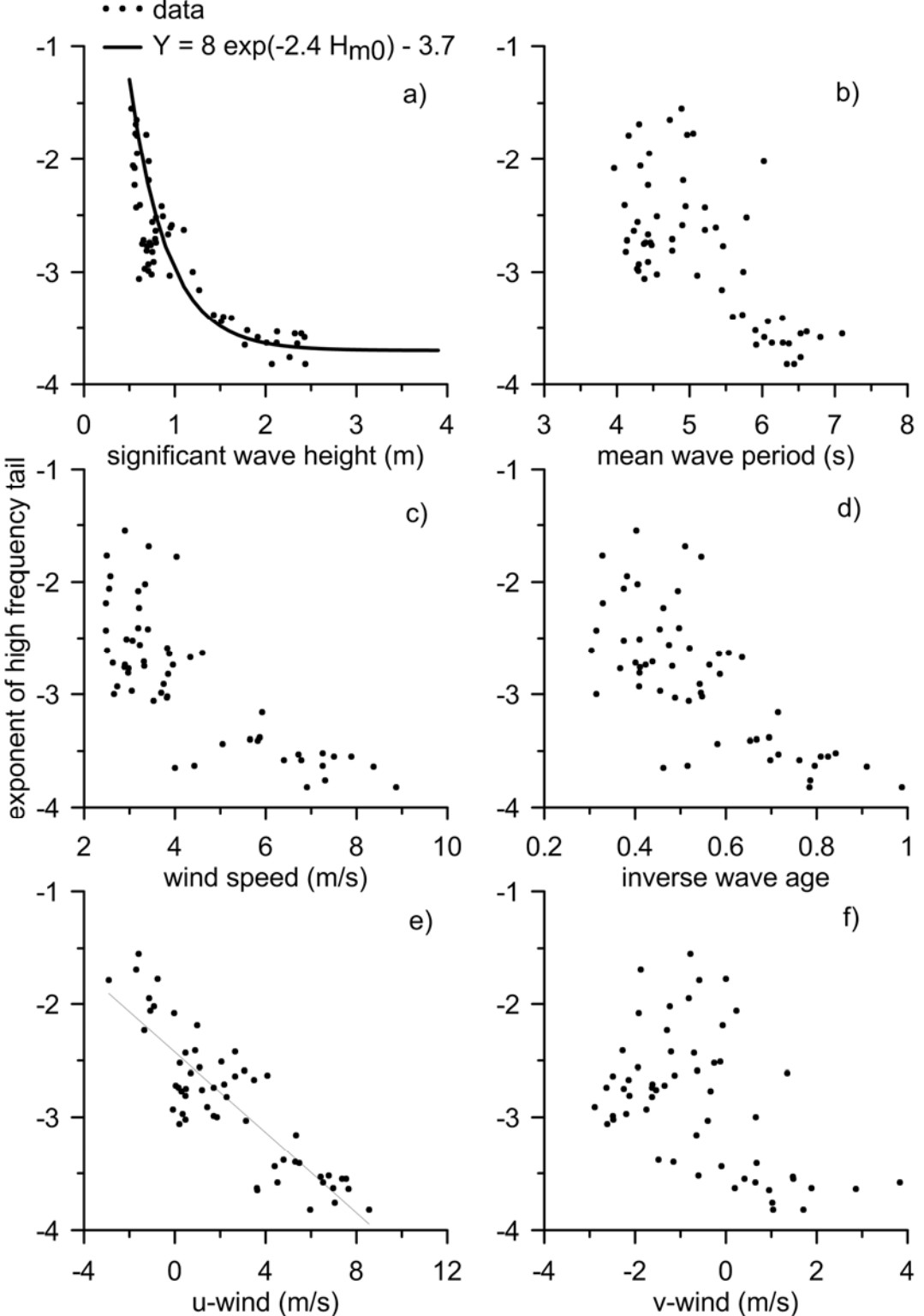

Figure 11. Plot of exponent of the high-frequency tail with a) significant wave height b) mean wave
period, c) wind speed, d) inverse wave age, e) u-wind and f) v-wind

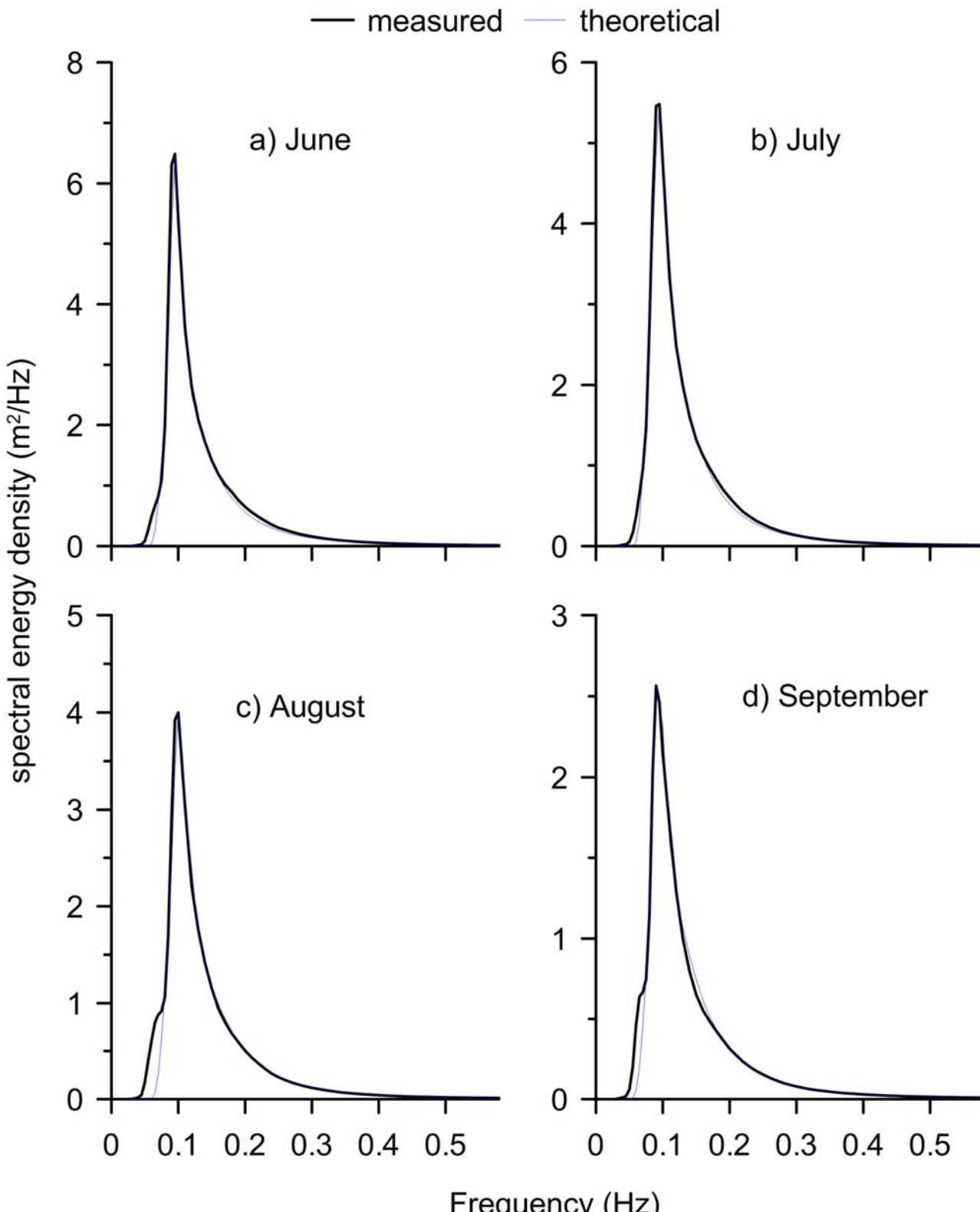

Figure 12. Fitted theoretical spectra along with the monthly average wave spectra for a) June, b)
July, c) August and d) September