# Peer review of "Wave spectral shapes in the coastal waters based on measured data off Karwar, west coast of"

_Ocean Science, 2016_

## Referee Comment (RC1) · Anonymous Referee #1 · 5 Feb 2017

General Comments

Generally, this work presents an impressive data set spanning 5 years. The data appear largely to be a new contribution to the literature, and if so, should qualify the work for publication as a case study. Having recognized that, there are several concerns with potential to change this assessment.

1) It appears that a significant portion of this data may have been published elsewhere. For example, here:

Glejin, J., Sanil Kumar, V., Amrutha, M.M. and Singh J., Characteristics of long-period swells measured in the in the near shore regions of eastern Arabian Sea, Int. J. Naval

Architecture and Ocean Engineering, 8, 312-319, 2016.

it seems clear that data from January 1 2011 through December 2012 were recorded at, or very near the location reported in this paper, with the same methods and perhaps instruments. The Glejin et al. paper present statistics that appear to overlap with statistics presented here.

2) Why is the broader-context not communicated? It seems peculiar that several existent publications by the authors use verbatim text describing the area, methods and motivation, as well as the same analysis techniques and products, but those works are referenced only narrowly in regard to specific details. For example, data collected over nearly the same period of record (March 23, 2010 to November 6, 2014) only a few kilometers up the coast, and with essentially the same analysis is reported here:

Anjali Nair, M. & Kumar, V.S., Spectral wave climatology off Ratnagiri, northeast Arabian Sea, Nat Hazards (2016) 82: 1565. doi:10.1007/s11069-016-2257-5

These publications share significant portions of text and techniques:

Glejin, J., Sanil Kumar, V., Sajiv, P.C., Singh, J., Pednekar, P., Ashok Kumar, K., Dora, G.U., and Gowthaman, R., Variations in swells along eastern Arabian Sea during the summer monsoon, Open J. Mar. Sci., 2 (2), 43–50, 2012.

Glejin, J., Sanil Kumar, V., Amrutha, M.M. and Singh J., Characteristics of long-period swells measured in the in the near shore regions of eastern Arabian Sea, Int. J. Naval Architecture and Ocean Engineering, 8, 312-319, 2016.

Indeed, there appears to be such a plethora of work in the area that a recent compendium was published:

P. V. Ethamony, R. Rashmi, S.V. Samiksha, V. M. Aboobackerm, Recent Studies on Wind Seas and Swells in the Indian Ocean: A Review, The International Journal of Ocean and Climate Systems, Vol 4, Issue 1, pp. 63 - 73.

Informing the reader of the broader context in which these measurements exist, and to clarify novel differences between published work and data presented in this paper is warranted.

3) The expressed objective of the work is: "This study addresses two main questions: (1) How the high-frequency tail of the wave spectrum varies in different months? (2) What are the spectral parameters for the best-fit theoretical spectra?"

The first objective is questionable since the method to assess slopes of the spectral tails is not revealed. The second objective is questionable since an ad hoc method consisting of a piecewise concatenation of different theoretical spectral functions is used. Specifics of how the concatenated frequency bands are determined is not provided, nor is an assessment of the physical assumptions inherent in concatenation of these different spectra.

The authors have attempted to clarify the data with statistical analysis, but the specific methods and algorithms are not provided, a prime deficiency. If there is desire to improve understanding and interpretation of the data, some effort to connect the data/statistics to underlying physical processes could improve the publication value. For example, you have argued that "Understanding of the wave spectral shapes is of primary importance for the design of marine facilities" yet little effort is made to quantify or relate physical parameters/forcings pertaining to the change in spectral slopes/peaks.

4) Please address the lack of consistency in precision of your statistical estimates and the lack of standard error or confidence limits. For example, wave frequencies are variously reported with precisions of 1, 2 or 3 decimal places, even in the same sentence. Slopes of the wave spectra are reported with 1 or 2 decimal point precision. Kindly adopt a uniform usage for expressing the precision of your estimates, ideally ones that reflect physically pertinent cutoffs. Please add error estimates or confidence bounds to your estimates, or address quantitatively why they are not germane.

5) You quote estimates of the slope of high-frequency portions of the wave spectra, but never define how those estimates are made.

6) You plot and describe "normalized" spectral densities, but never detail how this normalization is applied. One is then left to question whether the normalization is specific to each time period, or normalized across all periods so that direct comparisons of different years is meaningful.

7) Line 92 "The average monthly sea level at Karwar varies from 1.06 m (in September) to 1.3 m (in January)"

You are apparently quoting geodetic elevations here, but there is no reference datum specified.

8) Line 100 "The data for every 30 minutes from the continuous records at 1.28 Hz are processed as one record. From the time series data, the wave spectrum is obtained through fast Fourier transform (FFT)."

My understanding of waverider spectral processing is that motion samples are recorded at a sample rate of fs = 2.56 Hz, not 1.28 Hz. The bandwidth of the spectral estimate is fs/2 = 1.28 Hz. I do not believe that the data over 30 minutes are processed as one record. In standard Datawell processing, records of length 200 seconds are collected (N = 512 samples). 17 records are then FFT'd, windowed, and averaged with a 50% overlap to produce the 30 minute power spectral density estimate. Kindly verify your description, and if it differs from the standard Datawell spectral processing, detail and justify the differences.

9) Line 185 - 189 Here you describe dispersion relation impacts characterizing the wave spectrum of your data, you might consider moving this information to the beginning of the section.

10) Line 268 "Contour plots of spectral energy density (normalized) clearly show the predominance of wind-seas and swells during the non-monsoon period (Fig. 9)."

Here you have apparently applied the unspecified 'normalization' to month long temporal windows of spectral density averaged over years. Is the normalization value specific to each month, or is it over the entire ensemble so that meaningful intercomparisons can be made? The normalization, as well as the averaging must be defined.

11) Line 281 "To study the characteristics of different wave systems, average mean wave direction and average wave spectral energy density grouped under different peak frequency bins are plotted in Fig. 10."

The meaning of "wave systems" is not clear. The meaning of "average mean" is not clear. Please see the following comment.

12) Line 281 - 294 It isn't clear to me that this section along with figures 10 and 11 provide meaningful analysis that isn't already discernible from the previous results, rather this seems like part of the exploratory data analysis and seems redundant. My recommendation is to remove this section and figures.

13) Line 296 "The behavior of the high-frequency part of the spectrum is governed by the energy balance of waves generated by the local wind fields. When the wind blows over a long fetch or for a long time, the wave energy for a given frequency reaches the equilibrium range and the energy input from the wind are balanced by energy loss to other frequencies and by wave breaking."

You have argued that the change in slope is indicative of a change from local wind-dominated to swell waves, a physical connection. And have recognized that the dynamic equilibrium between generation and dissipation processes are what control this slope... but fail to make any meaningful physical connection between the slope behavior and the underlying physics. This is not required in a paper that only presents data, but you have invested some effort in quantifying the spectra, it would seem natural to attempt a physical connection to sea state, wave height, wave age, wind speed or some physical parameter.

14) Line 296 - 313 This section seems to try and convey in detail a very simple observation that more energetic wave spectra have steeper high-frequency tail slopes. If this needs explanation at all, it seems it should be much simpler. Again, how the slopes are computed is not defined.

Why are slopes in table 4 and figure 12 numerically negative, yet in the text are all positive? You refer to slopes as increasing, but are they not becoming more negative?

15) Line 307 "It is shown in Fig. 12 that the slope also increases as the mean wave period increases."

Figure 12 presents an interesting association between high frequency wave spectra slope and monthly mean wave height, but there is only one sentence referring to it with no exploration of its physical significance. It suggests a nonlinear saturation of slope as a function of wave height. You may wish to consider the suggestion following the comments.

16) Line 312 "The slope of the high-frequency end of the wave spectrum becomes milder when the wave nonlinearity increases. The study shows that the tail of the spectrum is influenced by the local wind conditions."

An attempt to physically connect your observations/statistics to physical forcing... Good. But this is unsatisfying. You are arguing that by virtue of large Hm0 alone that nonlinearity increases? This seems speculative at best.

17) Line 315 4.3 Theoretical wave spectra

This section is very unsatisfying. How the fits to the theoretical spectra are determined is not provided, the only clue is: 'values for $\alpha$ and Γ̌ were randomly varied within a range'. That this is not explained at all is a serious deficiency.

18) Line 317 "The monthly average wave spectra for the year 2015, is compared with JONSWAP and Donelan theoretical wave spectra."

Why was the only the year 2015 used in the monthly average wave spectra fits? Is there reason to believe that interannual variability is insignificant and can be ignored, figure 6 suggests not. How then can monthly means from one year be considered representative of the wave climate? Year 2015 is reported to have 14772 observations, while years 2011, 2012 and 2014 have 17300. How do you justify extracting mean statistics on a data set missing 15% of the data?

19) Line 320 - 326 "For these months the first peak is fitted with JONSWAP spectrum and second with Donelan, and the fitted spectrum shows a good match with the measured one. In the monsoon period, the spectrum is single peaked with high spectral energy density and during this period JONSWAP spectrum is fitted up to the peak frequency and after that Donelan spectrum is used. During the months May, October and November, after the peak frequency, the measured spectrum is not smooth and hence for this part, Donelan spectrum is fitted in two parts in order to obtain the best fit."

The ad hoc piecewise fitting of spectra is unsatisfying. A uniform criteria from which spectra are fit to different portions of the data does not seem to exist. How can an ad hoc scheme be deemed useful to those engaged in the "design of marine facilities"? How does one interpret the physical basis of these piecewise spectra? Does it make physical sense to apply an ad hoc piecewise spectral concatenation? Are the essential assumptions inherent in the different piecewise spectra being satisfied?

In absence of answering these questions, an alternative approach would be to fit the different theoretical spectra over the entire frequency band, present the results, and perhaps hypothesize/speculate why the different inherent assumptions in the theoretical spectra result in different fit fidelity over the different frequency bands/environmental conditions.

20) Figure 3 The color bands need improvement. For example, in a) there are two yellow bands with only slightly different saturation, but widely different amplitudes. It is difficult to discern which amplitude corresponds to the sections in the plot.

21) Figure 8 Instead of scaling the ordinal axis for each plot to maximize the dynamic range of the curves, it would be more informative to have a single uniform axis for all plots with the same angular range.

22) Figure 9 This is commonly referred to as a spectrogram. The amplitude scale has not been defined. No units are shown.

23) Figure 10 - recommend removing this figure and section To make figure 10 more informative for relative comparison across spectral bands, the ordinal axes should have uniform ranges, both for spectral amplitude and direction. This will allow the reader to immediately discern important differences between bands.

24) Figure 11 - recommend removing this figure and section Abcissal axes need labels and units.

25) Figure 12 Needs to explicitly label ordinates as slopes.

26) Figure 13 In contradiction to the text describing this figure, I'm unable to see where the Donelan spectrum was applied.

Suggested Analysis

As mentioned above, Figure 12 presents an interesting association between high frequency wave spectra slope and monthly mean wave height suggesting a nonlinear saturation of slope as a function of wave height.

The introduction contains a nice discussion of findings that quantify high-frequency slopes, both theoretically and experimentally, with substantial support that in your oceanographic setting that the expected high-frequency decay would be affine with $f^{-4}$. You mention in general terms how this decay represents an equilibrium between dissipation and energy input dominated by local winds, it may be useful to think about specific processes. For example Donelan et al. (2012) find that in addition to the $k^{-4}$ dissipation that swells modulate the equilibrium in breaking waves dependent on the mean surface slope, while Melville (1994) also quantified a relation between wave

packet slopes and dissipation rate. These results are specific to breaking waves, but one might expect similar relations between surface dynamics and dissipation rate for non breaking waves. If you do not find existing literature pertinent to non-breaking wave dissipation, then perhaps a functional representation of the data shown in figure 12 might be useful in revealing something about the physical connection, and at the very least would provide a predictive basis relating spectral slopes with mean wave heights as a basis for future research.

To that end, you might wish to fit a function of the form: A * exp( $\lambda$ Hm0 ) + s0, with initial parameters of A = 8, $\lambda$ = -2.4, s0 = -3.7 to the data of figure 12. This is exemplified with a subset of your data below. With optimized parameter values you will then have a functional representation of your spectral slopes based on Hm0. Presumably, Hm0 along with other parameters (wavelength, wind...) may lead you or others to a hypothesis relating your spectral slopes to sea surface physics.

[Figure]

[Figure]

[Figure]

**Fig. 1.** Slope vs Hm0

---

## Referee Comment (RC2) · Anonymous Referee #2 · 8 Mar 2017

Review of the manuscript os-2016-91: "Wave spectral shapes in the coastal waters based on measured data off Karwar, west coast of India" by Anjali Nair M., Sanil Kumar V.

The manuscript presents a compressive study of the wave regime on the west coast of India, based on in-situ buoy observations collected from a wave rider buoy, where the different spectral shapes, due to the monsoon regime in the area are studied. I find the study useful and interesting, and certainly worth of publication, pending on some queries below. The use of English is not very good. I have added some corrections, but there is only so much one can correct, hence the authors should consider revising the whole text carefully.

L15: erase "the". Wave spectra collected from where/what measuring system. Mention in the abstract that your study is based on buoy observations. L16: which coastal waters? L20: replace "no" with "not". L22: add "," after "period". L31: erase "the" L32: add "parameters" after "wave". L33: add "correspondent" before "spectral". L36: add "," after "decay". Replace "are" with "have been". L42: define JONSWAP acronym. Add "field campaign" after JONSWAP. L52 and L55: add "the" after "times". L60: add following reference: Ranjha, R., M. Tjernström, A. Semedo, G. Svensson, 2015: Structure and Variability of the Oman Coastal Low-Level Jet. Tellus A, 67, 25285, http://dx.doi.org/10.3402/tellusa.v67.2528 L61: define "AS" acronym. (It is defined latter in the text but it should be defined the first time it is used.) L67: You should state early in the text the monsoon, pre-monsoon, non-monsoon, etc. periods (time ranges in months) as soon as possible in the text. L75: remove "previous". Add "above" after "discussion". L76: measured how. State where the data is from. L77: add "," after "India". L79: replace "?" with ",". Add "and" before (2) L80: replace "?" with ",". L86: add "," after "north". L87: add "an icoming" after "have". L88: add "," before "since". L93 and L94: replace "tide" with "tides". L96: "Materials" or "Data"? L98 and L: replace "are" with "were". L100: sentence starting with "The data..."is confusing. Re-write. L101: add "a" before "fast". L105: add "," after "density". Replace "at" with "and f is the". Remove "f". L110: Consider using "U10" instead of "U". You will have to explain what you are using U10 for. You mention it here only, and then it seems that you do not use it, since it is never mentioned in the text. L136: regarding sentence starting with "In the monsoon...", where is this conclusion coming from? The sentence is also confusing, hence consider re-writing it. L146: replace "is" with "are". Wjat is ARB04? L148: sentence starting with "Small..." is confusing. Re-write. L152: replace "more" with "higher". L153: replace "frequency" with "frequencies". Replace "is" with "are". L154: erase "in". L156: add "and" before "intermediate". L168 and in some other parts of the text: Are you sure it is sea breeze. By definition sea breeze is (1) a rather local feature mid-afternoon feature, and (2) from the ocean to land. I am afraid this is not the case. Revise. L177: add "," before "hence". L178: add "the" before "monsoon". L179:

add "the" after "but". Add "from" before "SW". L180: add "," after "period". Replace "same" with "similar". L183: how do you know they are swells? L185: erase "of" and "to be". L193: normalized by what. Please provide additional information. L196: how do you separate the swells? Sentence starting with "Over an…" should be explained better. L203: replace "shifted" with "shifts". L212: why do you shift to frequency when you when talking about periods? L222: sentence starting with "Except…" is confusing. Re-write. L226: replace "the" with "wave", before "spectra". Add "that" before "the double". L227: add "frequent" before "during". L228: add "that" before "double". L 242: sentence starting with "Maximum…" is confusing. Re-write. L249: add "that" after "storm". L254: what do you mean by monsoon spectra? L255: I am afraid that the idea in the sentence starting with "Interannual…" cannot be stated like that. L263: replace "more" with "highest". L282: replace "plotted" with "shown". L299: replace "are" with "is". Replace "other" with "lower". L315: why this section (and the name of the section should be revised). What do you gain in adding this section? What exactly do you want to prove and add by adding the theoretical wave spectra. Where is the science here? You need to explain and defend this or else this section should be removed. L352: add "," after "investigated". Add "a" after "from".

---

## Author Comment (AC1) · 28 Mar 2017

Generally, this work presents an impressive data set spanning 5 years. The data appear largely to be a new contribution to the literature, and if so, should qualify the work for publication as a case study. Having recognized that, there are several concerns with potential to change this assessment.

1) It appears that a significant portion of this data may have been published elsewhere. For example, here: Glejin, J., Sanil Kumar, V., Amrutha, M.M. and Singh J., Characteristics of long-period swells measured in the in the near shore regions of eastern Arabian Sea, Int. J. Naval Architecture and Ocean Engineering, 8, 312-319, 2016. Reply: From the wave data collected for two years period (2011 and 2012) at the

study location, swells of period more than 18 s and significant wave height less than 1 m were separated and used to study the characteristics of low-amplitude long-period swells by Glejin et al. (2016). Glejin et al. (2016) presented the wave characteristics of low-amplitude long-period swells which occur for 1.4 to 3.6% of the time in a year. Statistics presented in Glejin et al. (2016) is different than that presented in this paper. Study on wave spectral shape and the interannual variations over a period of 5 years for this location has not been attempted so far. This information is now added in the revised paper. 2) Why is the broader-context not communicated? It seems peculiar that several existent publications by the authors use verbatim text describing the area, methods and motivation, as well as the same analysis techniques and products, but those works are referenced only narrowly in regard to specific details. For example, data collected over nearly the same period of record (March 23, 2010 to November 6, 2014) only a few kilometers up the coast, and with essentially the same analysis is reported here: Anjali Nair, M. & Kumar, V.S., Spectral wave climatology off Ratnagiri, northeast Arabian Sea, Nat Hazards (2016) 82: 1565. doi:10.1007/s11069-016-2257-5

These publications share significant portions of text and techniques: Glejin, J., Sanil Kumar, V., Sajiv, P.C., Singh, J., Pednekar, P., Ashok Kumar, K., Dora, G.U., and Gowthaman, R., Variations in swells along eastern Arabian Sea during the summer monsoon, Open J. Mar. Sci., 2 (2), 43–50, 2012.

Glejin, J., Sanil Kumar, V., Amrutha, M.M. and Singh J., Characteristics of long-period swells measured in the in the near shore regions of eastern Arabian Sea, Int. J. Naval Architecture and Ocean Engineering, 8, 312-319, 2016.

Indeed, there appears to be such a plethora of work in the area that a recent compendium was published:

P. Vethamony, R. Rashmi, S.V. Samiksha, V. M. Aboobacker M, Recent Studies on Wind Seas and Swells in the Indian Ocean: A Review, The International Journal of

Ocean and Climate Systems, Vol 4, Issue 1, pp. 63 - 73. Informing the reader of the broader context in which these measurements exist, and to clarify novel differences between published work and data presented in this paper is warranted.

Reply: Earlier the above publications were referred only at appropriate places in the manuscript. Now as per suggestion of the reviewer, we have added a paragraph in the introduction to provide a broader picture of the studies carried out in the eastern Arabian Sea and to clarify the differences between published work and the data presented in this paper.

3) The expressed objective of the work is: "This study addresses two main questions: (1) How the high-frequency tail of the wave spectrum varies in different months? (2) What are the spectral parameters for the best-fit theoretical spectra?"

The first objective is questionable since the method to assess slopes of the spectral tails is not revealed. The second objective is questionable since an ad hoc method consisting of a piecewise concatenation of different theoretical spectral functions is used. Specifics of how the concatenated frequency bands are determined is not provided, nor is an assessment of the physical assumptions inherent in concatenation of these different spectra.

The authors have attempted to clarify the data with statistical analysis, but the specific methods and algorithms are not provided, a prime deficiency. If there is desire to improve understanding and interpretation of the data, some effort to connect the data/statistics to underlying physical processes could improve the publication value. For example, you have argued that "Understanding of the wave spectral shapes is of primary importance for the design of marine facilities" yet little effort is made to quantify or relate physical parameters/forcings pertaining to the change in spectral slopes/peaks. Reply: We used the statistical curve fitting techniques to assess slopes of the spectral tail. Now we have added the below details.

An exponential curve $y = k.f^b$ is fitted for high frequency part of the spectrum and

[Figure]

the exponent (value of b) is estimated for the best fitting curve based on statistical measures such as least square error and bias. The slope of the high-frequency part of the wave spectrum is represented by the exponent of the high-frequency tail.

Specifics of how the concatenated frequency bands are determined is now added. Now we have deleted the representation of double-peaked wave spectrum with the theoretical spectrum.

For the present study JONSWAP spectrum is tested by fitting for the whole frequency range of the measured wave spectrum. It is found out that the JONSWAP spectra do not show good fit for higher frequency range, whereas Donelan spectrum shows better fit for high-frequency range. Hence, JONSWAP spectrum is used for lower frequency range up to spectral peak and Donelan spectrum is used for the higher frequency range from the spectral peak for single-peaked wave spectrum. Theoretical wave spectra is not fitted to the double-peaked wave spectra.

Now we have also added the plots showing the variation of high frequency tail with significant wave height, mean wave period, wind speed and inverse wave age.

4) Please address the lack of consistency in precision of your statistical estimates and the lack of standard error or confidence limits. For example, wave frequencies are variously reported with precisions of 1, 2 or 3 decimal places, even in the same sentence. Slopes of the wave spectra are reported with 1 or 2 decimal point precision. Kindly adopt a uniform usage for expressing the precision of your estimates, ideally ones that reflect physically pertinent cutoffs. Please add error estimates or confidence bounds to your estimates, or address quantitatively why they are not germane.

Reply: Consistency in precision is maintained now in the text. The wave spectrum is with a resolution of 0.005 Hz from 0.025 Hz to 0.1 Hz and is 0.01 Hz from 0.1 to 0.58 Hz. It is now added. 5) You quote estimates of the slope of high-frequency portions of the wave spectra, but never define how those estimates are made. Reply: Now it is explained as mentioned under 3.

6) You plot and describe "normalized" spectral densities, but never detail how this normalization is applied. One is then left to question whether the normalization is specific to each time period, or normalized across all periods so that direct comparisons of different years is meaningful.

Reply: Now it is added as below. Normalisation of the wave spectrum is done to know the spread of energy in different frequencies. Since the range of maximum spectral energy density in a year is large ($\sim$ 60 m2/Hz), each wave spectrum is normalised through dividing the spectral energy density by the maximum spectral energy density of that spectrum to understand the distribution of energy in different frequencies, specifically in the wind-sea and swell regions.

7) Line92 "The average monthly sea level at Karwar varies from 1.06 m (in September) to 1.3 m (in January)" You are apparently quoting geodetic elevations here, but there is no reference datum specified.

Reply: They are with respect to chart datum and is now added.

8) Line 100 "The data for every 30 minutes from the continuous records at 1.28 Hz are processed as one record. From the time series data, the wave spectrum is obtained through fast Fourier transform (FFT)."

My understanding of waverider spectral processing is that motion samples are recorded at a sample rate of fs = 2.56 Hz, not 1.28 Hz. The bandwidth of the spectral estimate is fs/2 = 1.28 Hz. I do not believe that the data over 30 minutes are processed as one record. In standard Datawell processing, records of length 200 seconds are collected (N = 512 samples). 17 records are then FFT'd, windowed, and averaged with a 50% overlap to produce the 30 minute power spectral density estimate. Kindly verify your description, and if it differs from the standard Datawell spectral processing, detail and justify the differences.

Reply: We used DWR-MKIII and the sampling interval is 3.84 Hz for this system. A
digital high-pass filter with a cut off at 30 s is applied to the 3.84 Hz samples. At the same time it converts the sampling rate to 1.28 Hz and stores the time series data at 1.28 Hz (Datawell, 2009). From the time series data for 200s, the wave spectrum is obtained through fast Fourier transform (FFT). During half an hour 8 wave spectra of a 200 s data interval each are collected and averaged to get a representative wave spectrum for half an hour (Datawell, 2009). These are now corrected in the paper. 2.56 Hz is correct for WR-SG. Now we have mentioned the type of buoy also to avoid confusion to the reader.

9) Line 185 - 189 Here you describe dispersion relation impacts characterizing the wave spectrum of your data, you might consider moving this information to the beginning of the section.

Reply: Moved to the beginning of the section

10) Line 268 "Contour plots of spectral energy density (normalized) clearly show the predominance of wind-seas and swells during the non-monsoon period (Fig. 9)."

Here you have apparently applied the unspecified 'normalization' to month long temporal windows of spectral density averaged over years. Is the normalization value specific to each month, or is it over the entire ensemble so that meaningful intercomparisons can be made? The normalization, as well as the averaging must be defined.

Reply: In Figure 9, each wave spectrum at 30 minutes interval is normalised through dividing the spectral energy density at each frequency by the maximum spectral energy density of that spectrum. Each normalised wave spectrum will have a maximum spectral energy density of 1. This is now explained in the paper.

Since the frequency bins over which the wave spectrum estimated is same in all years, the monthly and seasonally averaged wave spectrum is computed by taking the average of the spectral energy density at the respective frequencies of each spectrum over the specified time. Here normalisation is not done.

11) Line 281 "To study the characteristics of different wave systems, average mean wave direction and average wave spectral energy density grouped under different peak frequency bins are plotted in Fig. 10." The meaning of "wave systems" is not clear. The meaning of "average mean" is not clear.

Reply: Here the wave systems refers to wind-seas and swells approaching from different directions. Anyway this paragraph is now removed.

12) Line 281 - 294 It isn't clear to me that this section along with figures 10 and 11 provide meaningful analysis that isn't already discernible from the previous results, rather this seems like part of the exploratory data analysis and seems redundant. My recommendation is to remove this section and figures.

Reply: Figures 10 and 11 removed and the paragraph deleted.

13) Line 296 "The behavior of the high-frequency part of the spectrum is governed by the energy balance of waves generated by the local wind fields. When the wind blows over a long fetch or for a long time, the wave energy for a given frequency reaches the equilibrium range and the energy input from the wind are balanced by energy loss to other frequencies and by wave breaking."

You have argued that the change in slope is indicative of a change from local wind dominated to swell waves, a physical connection. And have recognized that the dynamic equilibrium between generation and dissipation processes are what control this slope... but fail to make any meaningful physical connection between the slope behaviour and the underlying physics. This is not required in a paper that only presents data, but you have invested some effort in quantifying the spectra, it would seem natural to attempt a physical connection to sea state, wave height, wave age, wind speed or some physical parameter.

Reply: We could not find a physical connection between the slope behaviour and the underlying physics. Slope is represented by the exponent of the high frequency tail.

[Figure]

We have added the scatter plot between exponent of the high frequency part of wave spectrum and significant wave height, mean wave period, wave age and wind speed.

14) Line 296 - 313 This section seems to try and convey in detail a very simple observation that more energetic wave spectra have steeper high-frequency tail slopes. If this needs explanation at all, it seems it should be much simpler. Again, how the slopes are computed is not defined.

Why are slopes in table 4 and figure 12 numerically negative, yet in the text are all positive? You refer to slopes as increasing, but are they not becoming more negative?

Reply: The methodology on computation of exponent of the high-frequency tail which represents the slope is now added.

The exponent is negative. As the exponent reduces, the slope of the spectral tail increases. Now it is corrected.

15) Line 307 "It is shown in Fig. 12 that the slope also increases as the mean wave period increases."

Figure 12 presents an interesting association between high frequency wave spectra slope and monthly mean wave height, but there is only one sentence referring to it with no exploration of its physical significance. It suggests a nonlinear saturation of slope as a function of wave height. You may wish to consider the suggestion following the comments.

Reply: Added the suggested analysis at the end.

16) Line 312 "The slope of the high-frequency end of the wave spectrum becomes milder when the wave nonlinearity increases. The study shows that the tail of the spectrum is influenced by the local wind conditions."

An attempt to physically connect your observations/statistics to physical forcing. Good. But this is unsatisfying. You are arguing that by virtue of large Hm0 alone that nonlin-

earity increases? This seems speculative at best.

Reply: Now we have added a Figure showing the variation of the skewness of the sea surface elevation data with the significant wave height to show that the nonlinearity increases with increase in Hm0. The below sentences are also added.

The most obvious manifestations of nonlinearity is sharpening of the wave crests and the flattening of the wave troughs and these effects are reflected in the skewness of the sea surface elevation (Toffoli, 2006). No skewness indicates linear sea states, positive skewness value indicate that the wave crests are bigger than the troughs. Figure 10 shows that nonlinearity increases with increase in Hm0.

17) Line 315 4.3 Theoretical wave spectra

This section is very unsatisfying. How the fits to the theoretical spectra are determined is not provided, the only clue is: 'values for $\alpha$ and ÏŠ were randomly varied within a range'. That this is not explained at all is a serious deficiency.

Reply: Explanation is given above for Qn. No 3. Equations used and parameters are also explained in Section 'Data and Methods'

18) Line 317 "The monthly average wave spectra for the year 2015, is compared with JONSWAP and Donelan theoretical wave spectra."

Why was the only the year 2015 used in the monthly average wave spectra fits? Is there reason to believe that interannual variability is insignificant and can be ignored, figure 6 suggests not. How then can monthly means from one year be considered representative of the wave climate? Year 2015 is reported to have 14772 observations, while years 2011, 2012 and 2014 have 17300. How do you justify extracting mean statistics on a data set missing 15% of the data?

Reply: There was no reason for selection of 2015. Now we have used 2011 during which 99.98% of data collected.

19) Line 320 - 326 "For these months the first peak is fitted with JONSWAP spectrum and second with Donelan, and the fitted spectrum shows a good match with the measured one. In the monsoon period, the spectrum is single peaked with high spectral energy density and during this period JONSWAP spectrum is fitted up to the peak frequency and after that Donelan spectrum is used. During the months May, October and November, after the peak frequency, the measured spectrum is not smooth and hence for this part, Donelan spectrum is fitted in two parts in order to obtain the best fit."

The ad hoc piecewise fitting of spectra is unsatisfying. A uniform criteria from which spectra are fit to different portions of the data does not seem to exist. How can an adhoc scheme be deemed useful to those engaged in the "design of marine facilities"?

How does one interpret the physical basis of these piecewise spectra? Does it make physical sense to apply an ad hoc piecewise spectral concatenation? Are the essential assumptions inherent in the different piecewise spectra being satisfied?

In absence of answering these questions, an alternative approach would be to fit the different theoretical spectra over the entire frequency band, present the results, and perhaps hypothesize/speculate why the different inherent assumptions in the theoretical spectra result in different fit fidelity over the different frequency bands/environmental conditions.

Reply: We tried fitting the spectra for the entire frequency range. Since it is not matching, we have fitted JONSWAP spectra in the low frequency range upto peak frequency and the Donelan spectrum for high frequency range from peak frequency. Now we deleted the fits of rdouble-peaked spectrum.

20) Figure 3 The color bands need improvement. For example, in a) there are two yellow bands with only slightly different saturation, but widely different amplitudes. It is difficult to discern which amplitude corresponds to the sections in the plot.

Reply: Color bands changed.

[Figure]

21) Figure 8 Instead of scaling the ordinal axis for each plot to maximize the dynamic range of the curves, it would be more informative to have a single uniform axis for all plots with the same angular range.

Reply: Now uniform axis is made for all plots with the same range.

22) Figure 9 This is commonly referred to as a spectrogram. The amplitude scale has not been defined. No units are shown.

Reply: The amplitude scale is 'normalised spectral energy density' and hence no unit.

23) Figure 10 - recommend removing this figure and section To make figure 10 more informative for relative comparison across spectral bands, the ordinal axes should have uniform ranges, both for spectral amplitude and direction. This will allow the reader to immediately discern important differences between bands.

Reply: Figure 10 removed.

24) Figure 11 - recommend removing this figure and section Abcissal axes need labels and units.

Reply: Figure 11 removed.

25) Figure 12 Needs to explicitly label ordinates as slopes.

Reply: Corrected in figure

26) Figure 13 In contradiction to the text describing this figure, I'm unable to see where the Donelan spectrum was applied.

Reply: Now it is explained as replied under 19.

Suggested Analysis

As mentioned above, Figure 12 presents an interesting association between high frequency wave spectra slope and monthly mean wave height suggesting a nonlinear saturation of slope as a function of wave height.

The introduction contains a nice discussion of findings that quantify high-frequency slopes, both theoretically and experimentally, with substantial support that in your oceanographic setting that the expected high-frequency decay would be affine with f -4. You mention in general terms how this decay represents an equilibrium between dissipation and energy input dominated by local winds, it may be useful to think about specific processes. For example Donelan et al. (2012) find that in addition to the k-4 dissipation that swells modulate the equilibrium in breaking waves dependent on the mean surface slope, while Melville (1994) also quantified a relation between wave packet slopes and dissipation rate. These results are specific to breaking waves, but one might expect similar relations between surface dynamics and dissipation rate for non breaking waves. If you do not find existing literature pertinent to non-breaking wave dissipation, then perhaps a functional representation of the data shown in figure 12 might be useful in revealing something about the physical connection, and at the very least would provide a predictive basis relating spectral slopes with mean wave heights as a basis for future research.

To that end, you might wish to fit a function of the form: A * exp( $\lambda$ Hm0 ) + s0, with initial parameters of A = 8, $\lambda$ = -2.4, s0 = -3.7 to the data of figure 12. This is exemplified with a subset of your data below. With optimized parameter values you will then have a functional representation of your spectral slopes based on Hm0. Presumably, Hm0 along with other parameters (wavelength, wind...) may lead you or others to a hypothesis relating your spectral slopes to sea surface physics.

Reply: Thanks for all the suggestions. We have added the above in the paper and revised.
* * *

---

## Author Comment (AC2) · 28 Mar 2017

All the suggested English language corrections carried out.

L42: define JONSWAP acronym. Add "field campaign" after JONSWAP. Reply: Added

L60: add following reference: Ranjha, R., M. Tjernström, A. Semedo, G. Svensson, 2015: Structure and Variability of the Oman Coastal Low-Level Jet. Tellus A, 67, 25285, http://dx.doi.org/10.3402/tellusa.v67.2528 Reply: Added the reference

L61: define "AS" acronym. (It is defined lat- ter in the text but it should be defined the first time it is used.) Reply: Defined

[Figure]

L67: You should state early in the text the monsoon, pre-monsoon, non-monsoon, etc. periods (time ranges in months) as soon as possible in the text. Reply: Now it is stated.

L76: measured how. State where the data is from. Reply: Modified.

L100: sentence starting with "The data. . ."is confusing. Re-write. Reply: Modified

L110: Consider using "U10" instead of "U". You will have to explain what you are using U10 for. You mention it here only, and then it seems that you do not use it, since it is never mentioned in the text. Reply: Now it is mentioned.

L168 and in some other parts of the text: Are you sure it is sea breeze. By definition sea breeze is (1) a rather local feature mid-afternoon feature, and (2) from the ocean to land. I am afraid this is not the case. Reply: Deleted.

L183: how do you know they are swells? Reply: Based on sea swell separation.

L193: normalized by what. Please provide additional information.

Reply: Now it is mentioned.

L196: how do you separate the swells? Sentence starting with "Over an. . ." should be explained better. Reply: It is added under data and methods. Now the sentence is corrected.

L212: why do you shift to frequency when you when talking about periods?

Reply: Now it is corrected.

L222: sentence starting with "Except. . ." is confusing. Re-write. Reply: Corrected.

L 242: sentence starting with "Maximum. . ." is confusing. Re-write. Reply: Corrected.

L254: what do you mean by monsoon spectra? Reply: It is the wave spectra averaged over monsoon period. Now it is corrected.

L255: I am afraid that the idea in the sentence starting with "Interannual. . ." cannot

be stated like that. Reply: Corrected as "Interannual variations within the spectrum are more for wind-sea region compared to swell region"

L315: why this section (and the name of the section should be revised). What do you gain in adding this section? What exactly do you want to prove and add by adding the theoretical wave spectra. Where is the science here? You need to explain and defend this or else this section should be removed.

Reply: We wanted to show that with suitable modification, we can fit a theoretical wave spectrum to the measured data. The coefficients obtained in fitting the spectrum are presented in Table 6.

Track-changed manuscript incorporating both the reviewers comments is attached as supplement.

Please also note the supplement to this comment:
http://www.ocean-sci-discuss.net/os-2016-91/os-2016-91-AC2-supplement.pdf

**Supplement:**

[revised manuscript text omitted]

**Most4.2 Wave spectrum**

219

233 234

235

| 236 | of the wave conditions (- 75%) are to be intermediate and shallow water waves at the buoy                     |
|-----|---------------------------------------------------------------------------------------------------------------|
| 237 | location (where water depth is less than half the wavelength, $d < L/2$ ), this condition is not satisfied    |
| 238 | during 25% of time due to waves with mean periods of 4.4 s or lessThis study, therefore, deals                |
| 239 | with shallow, intermediate and deepwater wave elimatology. Hence, bathymetry will significantly               |
| 240 | influence the wave characteristics,                                                                           |
| 241 | +                                                                                                      |
| 242 | Normalisation of the wave spectrum is done to know the spread of energy in different                          |
| 243 | frequencies. Since the range of maximum spectral energy density in a year is large (~ 60 m 2 /Hz), |
| 244 | each wave spectrum is normalised through dividing the spectral energy density by the maximum                  |
| 245 | spectral energy density of that spectrum. 4.2 Wave spectrum                                                   |

The normalized wave spectral energy density contours are presented for different years to 247 248 know the wind-sea/swell predominance (Fig. 5). The predominance of both the wind-seas and 249 swells are observed in the non-monsoon period, whereas in the monsoon only swells are predominant (Fig. 5). The separation of swells and wind-seas indicates that over<del>Over</del> an annual 250 cycle, around 54% of the waves are due to swells. Glejin et al. (2012) reported that the dominance 251 of swells during monsoon is due to the fact that even though the wind at the study region is strong 252 253 during monsoon, the wind over the entire AS also will be strong and when these swells are added to the wave system at the buoy location, the energy of the -swell increases (Donelan, 1987) and will 254 255 result in dominance of swells. The spread of spectral energy to higher frequencies (0.15 to 0.25 Hz) is predominant during January-May (Fig. 5) due to sea-breeze in the pre-monsoon period (Neetu et 256 al., 2006; Dora and Sanil Kumar, 2015). In the monsoon during the wave growth period, the 257 spectral peak shiftsshifted from 0.12-0.13 Hz to 0.07-0.09 Hz (lower frequencies). 258

246

259

An interesting phenomenon is that the long-period (> 18 s) swells are present for 260 261 2.5% of the time during the study period. The buoy location at 15 m water depth is exposed to waves from northwest to south with the nearest landmass at  $\sim 1500$  km in the northwest (Asia),  $\sim$ 262 263 2500 km in the west (Africa),  $\sim$  4000 km in the southwest (Africa) and  $\sim$  9000 
[revised manuscript text omitted]

343

358

344 To study the characteristics of different wave systems, average mean wave direction and 345 average wave spectral energy density grouped under different peak frequency bins are plotted in Fig. 10. Wave spectrum is mostly single peaked at low frequency bins and multi-peaked at higher 346 347 frequencies. Within the frequency range 0.07 to 0.15 Hz, the spectrum observed is a smooth single peaked spectrum, with high energy density. Maximum energy observed is within the frequency 348 349 range 0.08 0.1 Hz, having direction 240° which indicates the monsoon swells. Above 0.15 Hz, even though the spectrum is multi peaked, energy gradually decreases, with the lowest energy observed 350 between 0.3 and 0.5 Hz. Between 0.15 and 0.5 Hz, waves observed are from the northwest 351 direction ( $300^{\circ}$ ) and represents the wind seas produced by local winds. Long period swells (Tp > 352 14s) are also observed from the southwest. The average direction of waves with  $H_{m0} < 1m$  shows 353 the northwest wind seas and the southwest swells, whereas for high waves ( $H_{mh} > 3m$ ), the 354 difference between the swell and wind sea direction decreases (Fig. 11). This is because the high 355 waves get aligned to the bottom contour before 15 m water depth on its approach to the shallow 356 357 water.

359 The behavior of the high-frequency part of the spectrum is governed by the energy balance of waves generated by the local wind fields. When the wind blows over a long fetch or for a long 360 361 time, the wave energy for a given frequency reaches the equilibrium range and the energy input from the wind isare balanced by energy loss to lowerother frequencies and by wave breaking 362 363 (Torsethaugen and Haver, 2004). The high-frequency tail slope of the monthly average wave spectrum in different years (Table 4) shows that the slope is high (b < -(> -3.1)), during June to 364 365 September and the case is same for all the years studied (Table 4). During all other months, the exponent in the expression for the frequency tail slope-is within the range  $\frac{1.5}{1.5}$  3.1 to -1.5. The 366 367 distribution of exponentslope values for different significant wave height ranges shows that the slope increases (exponent decrease from -2.44 to -4.20) as the significant wave height increases 368 and reaches a saturation range. For frequencies from 0.230.229 to 0.58 Hz in the eastern AS during 369 January-May, Amrutha et al. (2017)(2016) observed that the high-frequency tail has  $f^{2.5}$  pattern at 370 15 m water depth and for frequencies ranging from 0.315 to 0.55 Hz, the high-frequency tail 371 follows f3 at 5 m water depth. It is shown in Fig. 12 that the slope also increases as the mean wave 372

373period increases, Since  $H_{mo}$  is maximum during the monsoon period, the slope is also maximum374during June to September. There is no much interannual variation in slope for swell dominated375spectra during the monsoon, while in the non-monsoon period when wind-seas have much376influence, the slope varies significantly. The slope of the high frequency end of the wave spectrum377becomes milder when the wave nonlinearity increases. The study shows that the tail of the378spectrum is influenced by the local wind conditions.

380 4.3 Theoretical wave spectra

379

381

382 The monthly average wave spectra for the year 2015, is compared with JONSWAP 383 and Donelan theoretical wave spectra. It is found that JONSWAP and Donelan spectra with modified parameters describe well the wave spectra at low frequencies and high frequencies 384 385 respectively. The most obvious manifestations of nonlinearity are sharpening of the wave crests and the flattening of the wave troughs and these effects are reflected in the skewness of the sea surface 386 387 elevation (Toffoli, 2006). Zero skewness indicates linear sea states, positive skewness value 388 indicate that the wave crests are bigger than the troughs. Figure 10 shows that nonlinearity 389 increases with increase in  $H_{m0}$ . The slope of the high-frequency end of the wave spectrum becomes 390 steeper when the wave nonlinearity increases. Donelan et al. (2012) find that in addition to the  $k^4$ 391 dissipation that swells modulate the equilibrium in breaking waves dependent on the mean surface 392 slope, while Melville (1994) also quantified a relation between wave packet slopes and dissipation 393 rate. These results are specific to breaking waves, but one might expect similar relations between 394 surface dynamics and dissipation rate for non breaking waves. A function of the form: A \* exp( $\lambda$ 395  $H_{m0}$ ) + s0, with initial parameters of A = 8,  $\lambda$  = -2.4, s0 = -3.7 is found to fit the exponent of the 396 high-frequency tail data with the significant wave height (Fig. 11a). The functional representation 397 of the exponent of the high-frequency tail data with  $H_{m0}$  shown in Fig. 11a might be useful in 398 revealing the physical connection, and at the very least would provide a predictive basis relating 399 spectral slopes with mean significant wave heights as a basis for future research. It is shown in Fig. 11b that the exponent decreases (slope increases) as the mean wave period increases. The study 400 401 shows that the tail of the spectrum is influenced by the local wind conditions (Fig. 11c) and the 402 influence is more with the zonal component (u) of the wind than on the meridional component (v) 403 (Figs. 11e and 11f). The exponent of the high-frequency tail decreases with the increase of the 404 inverse wave age  $(U_{10}/c)$ , where c is the celerity of the wave.

- - Formatted: Default

406 4.3 Comparison with theoretical wave spectra 407 408 From Fig. 13, we can see that spectrum is double peaked during January April and 409 December. For these months the first peak is fitted with JONSWAP spectrum and second with Donelan, and the fitted spectrum shows a good match with the measured one. In the monsoon 410 411 period, the spectrum is single peaked with high spectral energy density and during this period 412 JONSWAP spectrum is fitted up to the peak frequency and after that Donelan spectrum is used. 413 The monthly average wave spectra during the monsoon period for the year 2011, is compared with 414 JONSWAP and Donelan theoretical wave spectra in Figure 12. During the months May, October 415 and November, after the peak frequency, the measured spectrum is not smooth and hence for this 416 part, Donelan spectrum is fitted in two parts in order to obtain the best fit. Here also JONSWAP 417 shows best fit, up to the peak frequency. 418 419 It is found that JONSWAP and Donelan spectra with modified parameters describe well the wave-420 spectra at low frequencies and high frequencies respectively. The values for  $\alpha$  and  $\Upsilon$  were 421 randomly varied within a range to find out the values for which, the theoretical spectrum best fits 422 the measured spectrum and those values were used to plot the theoretical spectrum. The values of  $\alpha$ 423 and  $\Upsilon$  thus obtained, for June, July, August<del>each month</del> and Septemberthe range of frequencies for 424 which each spectrum is plotted are given in Table 6. From the table, the average values of  $\alpha$  and  $\Upsilon$ , for the monsoon months over 1 year period are obtained as 0.00090.0050 and 1.821.33 for 425 426 JONSWAP spectra and 0.02740.0254 and 1.641.27 for Donelan spectra respectively. These values 427 are less than the generally recommended values of  $\alpha$  and  $\Upsilon$ ; 0.0081 and 3.3.  $\alpha$  is a constant that is related to the wind speed and fetch length. For all the data, Donelan spectrum fitted is proportional 428 429 to  $f^n$ , where n is the exponent value of the high-frequency tail. From the table, it can be seen that  $\alpha$ has the highest values during the months June to August and is the same for both JONSWAP and 430 431 Donelan spectrum. This may be due to the high wind speed during monsoon. High values of  $\alpha$  were also observed during November, February and April, since these are the months during which sea 432 433 breeze has maximum influence, high  $\alpha$  values is due to the influence of local winds. There is no seasonal variation observed in the values of  $\gamma$ . For all the months, Donelan spectrum fitted is 434 proportional to f4, except during May and November, where spectrum is proportional to f2.8 and f 435

405

436 2.5, above frequencies 0.15 Hz and 0.28 Hz respectively. The theoretical spectrum JONSWAP and

[revised manuscript text omitted]

612 Stephen, F.B., and Tor Kollstad.: Field trials of the directional waverider, Proceedings of the First
 613 International Offshore and Polar Engineering Conference, Edinburgh, III 55–63, 1991.

- Torsethaugen, K., and Haver, S.: Simplified double peak spectral model for ocean waves, In:
  Proceeding of the 14th International Offshore and Polar Engineering Conference, 2004.
- 616 Vethamony, P., Rashmi, R., Samiksha, S.V. and Aboobacker, M.: Recent Studies on Wind Seas
- and Swells in the Indian Ocean: A Review, International J. Ocean and Climate Systems, 4, 63 73,
  2013.
- 619 Young, I.R. and Babanin, A.V.: Spectral distribution of energy dissipation of wind-generated
  620 waves due to dominant wave breaking, Journal of Physical Oceanography., 36(3), 376-394, 2006.
- Yuan, Y., and Huang, N.E.: A reappraisal of ocean wave studies, J. Geophys. Res.-Oceans.,
  117(C11), 2012.

Formatted: Don't adjust space between Latin and Asian text, Don't adjust space between Asian text and numbers, Tab stops: 2.5 cm, Left + 4.99 cm, Left + 5.49 cm, Left + 6 cm, Left + 6.99 cm, Left + 7.99 cm, Left + 8.99 cm, Left + 9.99 cm, Left + 10.99 cm, Left + 11.99 cm, Left

623

**624 Figure captions**

- Figure 1. Study area along with the wave measurement location in eastern Arabian Sea
- Figure 2. Time series plot of a) significant wave height, b) mean wave period, c) peak wave period
  and d) mean wave direction from 1 January 2011 to 31 December 2015. Thick blue line indicates
  the monthly average values
- Figure 3. Wave roses during 2011-2015 (a) significant wave height and mean wave direction, (b) peak wave period and mean wave direction, (c) percentage of swell, (d) percentage of wind-sea and mean wave direction
- Figure 4. Date verses year plot of a) significant wave height b) mean wave direction, c) peak waveperiod and d) mean wave period
- Figure 5. Temporal variation of normalized spectral energy density (top panel) and mean wavedirection (bottom panel) with frequency in different years
- 636 Figure 6. Monthly average wave spectra in 2011 to 2015
- Figure 7. Wave spectra averaged over a) pre-monsoon (February-May), b) monsoon (June-September), c) post-monsoon (October-January) and d) full year in different years
- 639 Figure 8. Monthly average wave direction at different frequencies in different months
- Figure 9. Temporal variation of normalized spectral energy density in different months (data from2011 to 2015 used)
- Figure 10. Scatter plotPlot of significantaverage spectral energy density and average mean wave
   height with skewnessdirection of the sea surface elevation inwaves grouped under different
   yearspeak frequency bins
- Figure 11. Plot of exponenta) average spectral energy density and b) average mean wave direction of the waves under different  $H_{m0}$  with frequency
- Figure 12. Plot of high\_-frequency tail with a) significant wave height, b) (left panel) and with
  mean wave period, c) wind speed, d) inverse wave age, e) u-wind and f) v-wind (right panel)
- Figure 12.13. Fitted theoretical spectra along with the monthly average wave spectra for different
   month

**Table 1. Number of data used in the study in different years along with range of significant wave**

**height and average value**

| 6 | 5 | Λ |
|---|---|---|
| σ | э | 4 |

| Year | Significant w | Significant wave height (m) |       | %of data |
|------|---------------|-----------------------------|-------|----------|
|      | Range         | Average                     |       |          |
| 2011 | 0.3-4.4       | 1.1                         | 17517 | 99.98    |
| 2012 | 0.3-3.7       | 1.1                         | 17323 | 98.61    |
| 2013 | 0.3-3.6       | 0.9*                        | 14531 | 82.94    |
| 2014 | 0.3-4.5       | 1.1                         | 17284 | 98.65    |
| 2015 | 0.3-5.0       | 1.1                         | 14772 | 84.32    |

655 \* average value is estimated excluding the JulyAugust month data

**Table 2. Characteristics of waves in different range of significant wave height**

| Significant wave height         | Number
(percentage) | Range of Tp
(s) | Mean Tp (s) | Range of $T_{m02}(s)$ | Mean T m02
(s) |
|---------------------------------|------------------------|--------------------|-------------|-----------------------|------------------------------|
| range                           | 52062 (63.94)          | 26222              | 12.2        | 27105                 | 1.9                          |
| $1 \le H_{m0} \le 1 \ \text{m}$ | 18297 (22.47)          | 3.6-22.2           | 10.5        | 3.4-10.7              | 5.7                          |
| $2 \le H_{m0} \le 3 m$          | 9839 (12.08)           | 6.2-18.0           | 10.8        | 5.0-8.9               | 6.5                          |
| $3 \le H_{m0} \le 4 m$          | 1096 (1.35)            | 10.0-14.3          | 11.8        | 6.1-9.1               | 7.2                          |
| $4 \text{ m} \le H_{m0}$        | 133 (0.16)             | 10.5-14.3          | 12.6        | 7.2-9.3               | 7.8                          |

Table 3. Average wave parameters and number of data in different spectral peak frequencies

| Frequency (f p ) | Number of data | H m0 | T m02 | Peak wave period |
|-----------------------------|----------------|-----------------|------------------|------------------|
| range (Hz)                  | and %          | (m)             | (s)              | (s)              |
| $0.04 < f_p \le 0.05$       | 318 (0.39)     | 0.73            | 5.24             | 20.19            |
| $0.05 < f_p \le 0.06$       | 5341 (6.56)    | 0.82            | 5.48             | 17.16            |
| $0.06 < f_p \le 0.07$       | 14764 (18.13)  | 0.75            | 5.22             | 14.73            |
| $0.07 < f_p \le 0.08$       | 18221 (22.38)  | 0.80            | 5.05             | 12.96            |
| $0.08 < f_p \le 0.10$       | 25364 (31.15)  | 1.55            | 5.76             | 10.88            |
| $0.10 < f_p \le 0.15$       | 9459 (11.62)   | 1.25            | 5.35             | 8.07             |
| $0.15 < f_p \le 0.20$       | 6355 (7.80)    | 0.76            | 4.43             | 5.72             |
| $0.20 < f_p \le 0.30$       | 1487 (1.83)    | 0.78            | 3.86             | 4.36             |
| $0.30 < f_p \le 0.50$       | 118 (0.14)     | 0.66            | 3.22             | 3.09             |

| 674 | Table 4. Exponent Slope of the highfrequency tail of the monthly average wave spectra in |
|-----|------------------------------------------------------------------------------------------|
| 675 | different years                                                                          |
| 676 |                                                                                          |
|     |                                                                                          |

| Months    |       | Exp   | onent of the hig                    | h- High frequ | ency tail |           |
|-----------|-------|-------|-------------------------------------|----------------------|-----------|-----------|
| Monuis    | 2011  | 2012  | 2013                                | 2014                 | 2015      | 2011-2015 |
| January   | -2.08 | -2.93 | -2.97                               | -2.72                | -2.81     | -2.72     |
| February  | -2.41 | -3.02 | -2.74                               | -2.99                | -3.06     | -2.85     |
| March     | -2.75 | -2.91 | -2.82                               | -2.76                | No data   | -2.81     |
| April     | -2.56 | -2.74 | -2.64                               | -2.71                | -2.19     | -2.60     |
| May       | -2.59 | -2.67 | -2.63                               | -2.42                | -2.51     | -2.56     |
| June      | -3.64 | -3.53 | -3.55                               | -3.82                | -3.58     | -3.55     |
| July      | -3.76 | -3.55 | No data -
<del>3.40</del> | -3.82                | -3.63     | -3.70     |
| August    | -3.63 | -3.58 | -3.40 No
data             | -3.52                | -3.65     | -3.58     |
| September | -3.41 | -3.44 | -3.16                               | -3.38                | -3.00     | -3.30     |
| October   | -2.02 | -2.77 | -3.03                               | -2.52                | -2.61     | -2.68     |
| November  | -1.78 | -2.43 | -1.77                               | -1.55                | -1.65     | -1.84     |
| December  | -1.69 | -2.23 | -1.95                               | -2.06                | -1.79     | -1.94     |

Table 5. Exponent of the high-High-frequency tail of the average wave spectra in different wave height ranges

| Range of $H_{m0}(m)$ | Exponent of the high-High            |
|----------------------|--------------------------------------|
|                      | frequency tail <del>-parameter</del> |
| 0-1                  | -2.44                                |
| 1-2                  | -3.26                                |
| 2-3                  | -3.67                                |
| 3-4                  | -4.21                                |
| 4-5                  | -4.21                                |

Formatted Table Formatted: Centered

| 689 | Table 6. Parameters of the fitted wave spectrum in different yearsmonths |
|-----|--------------------------------------------------------------------------|
| 690 |                                                                          |
| 691 |                                                                          |
| 1   |                                                                          |

| Year        |               | JONSV          | JONSWAP spectrum |               | Donelan spectrum |  |
|-------------|---------------|----------------|------------------|---------------|------------------|--|
|             |               | α              | Ϋ́               | α      | Ϋ́               |  |
| 2011 | June          | 0.0013  | 2.2       | 0.0028 | 2.0       |  |
|             | July   | 0.0016  | 1.5       | 0.0021 | 1.7       |  |
|             | August        | 0.0013  | 1.8       | 0.0029 | 1.7       |  |
|             | September     | 0.0004  | 2.3       | 0.0021 | 1.6       |  |
| 2012 | June          | 0.0015  | 1.6       | 0.0029 | 2.0       |  |
|             | July          | 0.0010  | 2.1       | 0.0031 | 1.9       |  |
|             | August        | 0.0009  | 2.2       | 0.0032 | 1.7       |  |
|             | September     | 0.0006  | 2.0       | 0.0024 | 1.8       |  |
| 2013 | June          | 0.0006  | 3.3       | 0.0030 | 1.9       |  |
|             | July   | No data |                  |               |                  |  |
|             | August        | 0.0012  | 1.1       | 0.0038 | 1.4       |  |
|             | September     | 0.0005  | 1.9       | 0.0042 | 1.4       |  |
| 2014 | June          | 0.0010  | 1.1       | 0.0010 | 1.6       |  |
|             | July   | 0.0006  | 2.5       | 0.0019 | 1.2       |  |
|             | August | 0.0006  | 1.5       | 0.0021 | 1.2       |  |
|             | September     | 0.0011  | 1.1       | 0.0032 | 1.4       |  |
| 2015 | June   | 0.0011  | 1.4       | 0.0023 | 1.8       |  |
|             | July          | 0.0011  | 1.9       | 0.0024 | 1.8       |  |
|             | August        | 0.0008  | 1.8       | 0.0024 | 1.4       |  |
|             | September     | 0.0006         | 1.3       | 0.0043 | 1.6       |  |

Figure 1. Study area along with the wave measurement location in eastern Arabian Sea697

---

## Author Response (AR2)

**Reply to reviewer 1:**

Line 74
"This study shows that the percentage of swells in the measured waves was 75 to 79% at the locations with higher percentage of swells in the northern portion of AS compared to that at the southern side."

-- Unclear, please restate.

**Reply:** Corrected as below.

This study shows that the percentage of swells in the measured waves was 75% at the southern part of AS and 79% at the northern part of AS.

Line 112
-- Chart datum is ambiguous. A chart datum is likely a mean-low-lower-water or some other tidal datum. A tidal datum is not a geodetic datum. In this era of sea level-rise consciousness, it would be nice to have the geodetic water levels quantified, but you seem unwilling to invest the effort. I don't see the advantage of belabouring this point since it's the relative change in sea level that is apparently the point of this text. Why not just avoid the technicality of understanding what a geodetic datum is, which one applies, the needed transformations from a chart or tidal datum, and just say that there is a 24 cm annual cycle in mean sea level from September to January.

**Reply:** Corrected as per suggestion. Now it is as below.

There is a 0.24 m annual cycle in mean sea level from September to January.

Line 159
"An exponential curve y = k.f^b is fitted for high-frequency part of the spectrum"

-- what is k.f? Apparently k is a coefficient?

**Reply:** Corrected as kf^b. It is mentioned that k is coefficient.

Line 236
"Since the range of maximum spectral energy density in a year is large (~ 60 m^2 /Hz), each wave spectrum is normalised through dividing the spectral energy density by the maximum spectral energy density of that spectrum."

-- So you are 'normalizing' each spectra independently. Why would one do this? This complicates the utility of the spectral amplitude since each spectra has it's own unique scaling and cannot be directly compared to another spectra, nor can actual water level amplitudes be recovered from the spectra without the specific scaling.

When you say that "each wave spectrum is normalised through dividing the spectral energy density by the maximum spectral energy density of that spectrum", are you literally normalizing each individual spectrum, or are you normalizing the entire spectrogram over each year? The former is unacceptable, the latter imposes that year-to-year amplitude comparisons must be scaled.

To the point of spectral amplitudes, you note that wave spectra have physical units (m^2/Hz), thereby one can obtain an RMS wave height at any frequency if the spectral amplitude is known. Clearly that has been obviated by your normalization and the amplitude scale in figure 5 of [0,1].

The reader is not in a position to know how the spectral normalization is applied in time and to what extent individual spectra can be compared to each other, or that a physical measurement can be extracted from the data presented in figure 5. One of the strengths of this paper is its data coverage and analysis. To remove the ability for the reader to extract a physical amplitude from the data, or to have a uniform comparison across time is difficult to understand.

These same comments/issues apply to figure 9 and the monthly mean spectral amplitudes.

If one desires to address that "the range of maximum spectral energy density in a year is large", a conventional approach is express the spectral power in decibels.

**Reply:** Now we have presented the maximum spectral energy density used for normalising each wave spectrum in Figure 2e.

In the caption of Figure 5 and 9, the below sentence is also added.

The value used for normalizing the spectral energy density is presented in Fig. 2e.

The wave spectrum is normalised and presented only to show the predominance of wind-seas or swells. Only Figs. 5 and 9 present the normalised spectral energy density. Remaining wave spectra are not normalised ones. This is now mentioned in line 326.

Line 379
"The values for $\alpha$ and $\gamma$ were randomly varied within a range to find out the values for which, the theoretical spectrum best fits the measured spectrum and those values were used to plot the theoretical spectrum."

-- This issue remains unanswered. You have not specified what "randomly varied" and "over a range" mean. If your work is to be reproducible, should not you reveal the methods?

**Reply:** Now the range is given as below.

The values for $\alpha$ and $\Upsilon$ were varied from 0.0001 to 0.005 and 1.1 to 3.3 respectively to find out the values for which, the theoretical spectrum best fits the measured spectrum and those values were used to plot the theoretical spectrum.

We thank the reviewer for these comments. Now we have addressed these points in the revised manuscript. The changes are shown in the attached manuscript.

[revised manuscript text omitted]